## Replication

neuroscience/computational biology

network neuroscience, connectomics, replication, Human Connectome Project, adolescent brain cognitive development, functional connectivity

**Author for correspondence:**
Dustin Moraczewski
e-mail: dmoracze@nih.gov

†These authors contributed equally to this work.

# The positive–negative mode link between brain connectivity, demographics and behaviour: a pre-registered replication of Smith et al. (2015)

Nikhil Goyal[1,†], Dustin Moraczewski[1,†], Peter A. Bandettini[2], Emily S. Finn[2,3] and Adam G. Thomas[1]

[1]Data Science and Sharing Team, and [2]Section on Functional Imaging Methods, National Institute of Mental Health, Bethesda, MD, USA
[3]Department of Psychological and Brain Sciences, Dartmouth College, Hanover, NH, USA

NG, 0000-0002-5836-9244; DM, 0000-0002-0422-3135; PAB, 0000-0001-9038-4746; ESF, 0000-0001-8591-3068; AGT, 0000-0002-2850-1419

In mental health research, it has proven difficult to find measures of brain function that provide reliable indicators of mental health and well-being, including susceptibility to mental health disorders. Recently, a family of data-driven analyses have provided such reliable measures when applied to large, population-level datasets. In the current pre-registered replication study, we show that the canonical correlation analysis (CCA) methods previously developed using resting-state magnetic resonance imaging functional connectivity and subject measures (SMs) of cognition and behaviour from healthy adults are also effective in measuring well-being (a 'positive–negative axis') in an independent developmental dataset. Our replication was successful in two out of three of our pre-registered criteria, such that a primary CCA mode's weights displayed a significant positive relationship and explained a significant amount of variance in both functional connectivity and SMs. The only criterion that was not successful was that compared to other modes the magnitude of variance explained by the primary CCA mode was smaller than predicted, a result that could indicate a developmental trajectory of a primary mode. This replication establishes a signature neurotypical relationship between connectivity and phenotype, opening new avenues of research in neuroscience with clear clinical applications.

# 1. Introduction

## 1.1. Background

Understanding how brain functional connectivity is linked to human behaviour is a major goal of neuroscience for its potential to elucidate the mechanisms of the brain that lead to human cognition [1]. By modelling functional magnetic resonance imaging (fMRI) data as a region-to-region functional network, researchers can leverage mathematical tools from the study of complex systems (e.g. graph theory) to probe these high-dimensional relationships [2–5].

## 1.2. Original study by Smith *et al.* (2015)

In their landmark 2015 study, Smith *et al.* [6] investigated the high-dimensional relationship between functional connectivity and behavioural and phenotypic measures using data on young adults aged 22–35 from the Human Connectome Project (HCP) [7]. To study the relationship between these subject measures (SMs) and functional connectomes, the authors calculated a 200-dimension group independent component analysis (ICA) functional parcellation for the 461 HCP subjects in their study. From this parcellation, they derived subject-level functional connectomes, which are symmetric matrices of the edge weights between all 200 nodes. Of the SM data available in the HCP dataset, Smith *et al.* chose 158 measures that were quantitatively and qualitatively appropriate for analysis.

Using canonical correlation analysis (CCA), which can simultaneously consider two variable sets from different modalities to uncover essential hidden and high-dimensional associations between the sets [8], the authors discovered that one statistically significant CCA mode (the primary mode) explained more of the observed covariance between functional connectivity and SMs compared to the other CCA modes. The authors found a strong positive correlation ($r = 0.87$, $p < 10^{-5}$) between the primary CCA mode connectome weights and SM weights. From the primary CCA mode, they derived a 'positive–negative axis' that quantified the correlation of various SMs to the CCA mode, linking lifestyle, demographic and psychometric SMs to each other and to a specific pattern of brain connectivity [6]. SMs commonly considered to be positive qualities (e.g. higher income, greater education level and high performance on cognitive tests) were positively correlated with the primary CCA mode, and SMs commonly considered to be negative qualities (e.g. substance use, rule-breaking behaviour and anger) were negatively correlated with the CCA mode.

A hierarchical clustering analysis of the 200 connectome nodes revealed clusters of nodes in four regions: one cluster in the sensory, motor, insula and dorsal attention regions; and three clusters covering the default mode network (DMN), subcortical and cerebellar regions [6]. The brain areas that contributed most strongly to variations in connectivity were similar to those associated with the DMN, suggesting that functional connectivity within (and, to some degree, between) the DMN may be important for higher-level cognition and behaviour [8].

Finally, to test the predictive value of their model, the authors conducted a train-test split in which 80% of subject data (randomly selected, respecting family relationships) were used to train a CCA model, which was then tested on the remaining 20% of the data. This process was repeated 10 times, with statistical significance estimated via permutation testing (1000 permutations) of family structures in each iteration. This train-test process yielded a statistically significant mean correlation of $r = 0.25$ between the primary CCA mode connectome and SM weights in the testing dataset, strongly supporting the primary CCA mode finding [6].

The findings from Smith *et al.* [6] suggest that high-dimensional patterns of positive and/or negative SMs can predict high-dimensional patterns of brain connectivity and vice versa. If one (SM or brain connectivity) can be used to predict the other, researchers move one step closer toward understanding the complex relationship between brain, lifestyle and behaviour [9]. Establishing a baseline neurotypical signature for this relationship will have important implications for understanding sub-clinical, clinical populations and other populations (e.g. developmental).

As a precursor to the current study, using code published by the original authors, we computationally replicated their findings in both the HCP 500 and 1200 subject releases to validate our own analysis pipeline [10].

## 1.3. Conceptual replication in an independent dataset

In the current study, we aim to conceptually replicate Smith *et al.*'s positive–negative axis and primary CCA mode findings in novel data from the Adolescent Brain Cognitive Development (ABCD) study.

This is not a replication in the sense that we seek to find results identical to those of the original authors (an exact computational replication of the original study is documented in our preprint [10]). Rather, we apply the methodology of Smith *et al.* to the ABCD dataset to determine if there exists a strong correlation between connectomes and SMs and a dominant CCA mode that explains a significant amount of covariance within a dataset with very different subject and scanner characteristics. The longitudinal ABCD study is an ongoing effort to study the environmental influences on behavioural and brain development in pre-adolescents, recruiting and following over 11 000 children aged 9–10 over a period of 10 years [11]. This multi-site study leverages multiple imaging modalities, including resting-state fMRI, to characterize brain development, providing a large neuroimaging dataset for exploration using the tools of network neuroscience.

We selected the ABCD dataset to replicate Smith *et al.* for multiple reasons. The 22- to 35-year-old subjects in the HCP dataset, which was fairly homogeneous in terms of subject demographics and scan quality [7], are vastly different from the pre-adolescents in the ABCD dataset in terms of brain development and demographics. Thus, this dataset presents a tremendous opportunity to search for the presence of a positive–negative axis in a heterogenous set of subjects who are still in neurodevelopmental stages. Additionally, the ABCD dataset is one of the largest available that has comparable SMs to the HCP dataset; since the original study was only conducted using 461 subjects, this increases the statistical power of the analysis. A study by Xia *et al.* used neuroimaging data of youth (8–22 years old) and found positively correlated patterns of functional connectivity and psychiatric symptoms across four dimensions of psychopathology. Each dimension was associated with a distinct pattern of abnormal connectivity while still showing common features (e.g. loss of network segregation) distinguishing them from normal brain function. To account for potential confounds, the authors controlled for age, sex, race and motion and they also found that these findings replicated in an independent dataset [12]. The implications of their study are threefold: (i) it is possible to perform a CCA-type analysis on highly heterogeneous youth data and obtain interpretable results; (ii) the analysis is sensitive enough to distinguish relationships among groups of related SMs and functional connectomes; and (iii) there appears to be a common neurobiological mechanism underlying vulnerability to a wide range of psychiatric symptoms, echoing the positive–negative axis finding. Given their findings, we felt that the ABCD dataset was appropriate for the current study despite the age difference and heterogeneity compared to HCP data.

Finding a primary CCA mode and positive–negative axis in the ABCD data would have implications beyond speaking to the robustness of the original study and the presence of the phenomena in a dataset other than HCP. It would suggest that the high-dimensional relationship between brain connectivity and phenotype may be independent of age, demographics, scanner type, preprocessing methods and even the SMs, which can vary in how they index cognitive constructs. Such observations would be pivotal for the establishment of a signature neurotypical relationship between connectivity and phenotype, which could then be applied to a variety of clinical populations potentially increasing its practical application in medicine. Most importantly, finding the positive–negative axis in the baseline ABCD dataset would enable researchers to study it longitudinally over subsequent ABCD releases for the next 10 years. In doing so, researchers could explore how the positive–negative axis changes with regard to age, behavioural characteristics, mental and physical health outcomes, education, demographics and environment, and possibly even determine which factors form the basis for the positive–negative axis. We aim to design the analysis pipeline such that it can easily be re-run by other researchers who wish to explore how these factors impact the CCA results over time as subsequent ABCD datasets are released. Given these potential implications, we feel that replicating the positive–negative axis in a heterogeneous dataset is important not only to lend validity to the original study but also to open new avenues of research in neuroscience.

### 1.3.1. Replication overview

In this conceptual replication, we followed the same methodology as Smith *et al.* to perform a CCA on ABCD subject connectomes and SM data (for full details, refer to §2). After applying similar preprocessing steps to the resting-state scan data, we derived a 200-dimension group-ICA functional parcellation. This parcellation was used to calculate subject-level connectomes, which were aggregated to form the connectome matrix used in the CCA. The choice of parcellation can have significant effects on connectivity results [13]. For replication purposes we chose to identify functional networks using the same 'soft' (e.g. data-derived spatial ICA) parcellation scheme as Smith *et al.*, which has

been shown to have greater predictive power compared to *a priori* functional and anatomical parcellations [5,14].

The ABCD study has over 60 000 SMs (this includes items from individual questions in an assessment to scan quality control (QC) measures), which we quantitatively filtered as per the original study. After removing quantitatively inappropriate SMs, we performed a one-to-one matching of ABCD SMs to their counterparts in the HCP dataset. This process identified 89 ABCD SMs, of which 74 were deemed appropriate for the CCA. We regressed out confounds similar to those in the original study prior to conducting the CCA.

After conducting the CCA, we performed multiple *post hoc* analyses from the original study, including generating a positive–negative axis that correlates the SMs (all of those included in the CCA, and four which were not) with the primary CCA mode; a hierarchical analysis of the connectomes, and determining which functional connections correlate most strongly with the primary CCA mode; a clustering analysis to identify the major brain regions into which the 200 nodes fall; and a train-test split to evaluate the predictive performance of the CCA model on unseen data.

Our criteria for a successful replication of the primary CCA mode (i.e. the strongest mode) were as follows:

(1) The primary CCA mode explains a statistically significant amount of variance in the connectomes and SMs relative to the null distribution generated via permutation testing.
(2) The primary CCA mode z-scores for connectomes and SMs are at least a factor of 2 and 3 greater, respectively, than the next largest z-scores for connectomes and SMs (from any of the other modes).
(3) A statistically significant correlation exists between the primary CCA mode SM and connectome weights ($p < 0.001$).

These criteria attempt to account for the large differences in the ABCD and HCP datasets by focusing on conceptually replicating the primary finding of the original study in the novel dataset, rather than replicating specific numerical results exactly. Although we are particularly interested in finding a single dominant CCA mode, given the large sample size of the ABCD dataset, it is possible that multiple significant modes exist. The appearance of these modes would not necessarily indicate a failure to replicate but would warrant a careful analysis of the results since in PCA/CCA there can be an arbitrary rotation among components that can spread the original result validly across the multiple significant modes [8]. Since the positive–negative axis, hierarchical analysis and train-test split were conducted *post hoc* in the original study, there was no definition for what constituted a successful replication for these aspects of the study.

# 2. Methods

This article received in-principle acceptance (IPA) at *Royal Society Open Science*. Following IPA, the accepted Stage 1 version of the manuscript, not including results and discussion, was pre-registered on the OSF (http://doi.org/10.17605/OSF.IO/HMC35). Minor deviations from the protocol are identified in footnotes, and detailed explanations of deviations are provided in §3.3.

## 2.1. Participants

The sample composition of the ABCD Release 2.0.1 before and after filtering out subjects is shown in table 1. The filtering process for excluding subjects is discussed in §2.3. Prior to excluding subjects, the sample contained a total of 11 875 pre-adolescents between ages 9 and 10. After filtering, the sample contained 7810 subjects.[1]

The ABCD dataset contained multiple types of sibships (children with the same parents): monozygotic (MZ) and dizygotic (DZ) twins, as well as siblings and single children. As in the original study, the sibship data were used to generate permutations of the ABCD SM data for statistical testing. Twins were designated as MZ or DZ based on genomic data, and, when zygosity was unavailable, twins were simply treated as siblings. There were 315 subjects that were designated as twins ($n = 285$) or triplets ($n = 30$) but were missing zygosity data; these subjects were considered non-twin siblings for our study to match the procedure followed by Smith *et al*. Prior to filtering, there were 738 MZ, 1082 DZ, 1908 non-twin siblings (1593 non-twin siblings plus 315 twins/triplets treated as non-twin siblings) and 8147 single children.

---

[1]Due to changes in the subject filtering criteria used in the final analysis, the revised sample size was 5013. See §3.3 for details.

**Table 1.** A breakdown of the ABCD demographics before and after filtering (outlined in §2.3). * = data not available for all subjects, percentages of available data are reported.

| | before filtering | after filtering[21] |
|---|---|---|
| no. subjects | 11 875 | 7810 |
| sex* | | |
| no. male | 6188 (52.1%) | 3894 (49.9%) |
| no. female | 5681 (47.9%) | 3914 (50.1%) |
| age in years (mean, std dev, range) | 9.91 | 9.96 |
| | 0.622 | 0.622 |
| | 9.0–10.92 | 9.0–10.92 |
| race/ethnicity* | | |
| white | 6174 (52.1%) | 4375 (56.1%) |
| black | 1779 (15.0%) | 1016 (13.0%) |
| Hispanic | 2407 (20.3%) | 1466 (18.8%) |
| Asian | 252 (2.1%) | 138 (1.8%) |
| other | 1245 (10.5%) | 805 (10.3%) |
| sibships (no. of subjects in each category) | MZ: 738 | MZ: 562 |
| | DZ: 1082 | DZ: 777 |
| | non-twin siblings: 1908 | non-twin siblings: 1245 |
| | single children: 8147 | single children: 5226 |

After filtering, there were 219 subjects that were designated as twins ($n = 197$) or triplets ($n = 22$), but were missing zygosity data; again, these subjects were considered as non-twin siblings for our study. The final sibship counts were: 562 MZ, 777 DZ, 1245 non-twin siblings (1026 non-twin siblings plus 219 twins/triplets treated as non-twin siblings) and 5226 single children.

## 2.2. Data acquisition

### 2.2.1. Imaging data

The ABCD functional MRI scans were acquired at 21 sites across the United States, using 3 T scanners from three manufacturers: Siemens, Philips and GE. The fMRI acquisition parameters were designed to be as similar as possible across all scanners: $90 \times 90$ matrix, 60 slices, spatial resolution 2.4 mm isotropic, TR = 0.8 s, multiband acceleration = 6 and multiband echo planar imaging. Subjects were scanned four times in 5-min scan lengths for a total of 20 min of resting-state scan time. T1- and T2-weighted scans with 1.0 mm isotropic resolution were also acquired and used for co-registration purposes. See [15] for more information on the ABCD fMRI scanning protocol.

### 2.2.2. Subject measure data

The ABCD SM battery includes a wide range of assessments of imaging data, biomarkers, cognitive function, substance use and abuse, mental and physical health and the youth's family and environment [16–19]. Given the longitudinal nature of the ABCD study, the battery was designed such that researchers can characterize subjects' changes in key domains over time. Measures were selected if they could be used through early adulthood or had parallel versions which were appropriate for older ages without their results becoming invalid over time due to repeated assessments or practice effects. A comprehensive computerized battery was administered at baseline during an in-person meeting; shorter follow-up assessments were administered over the phone at six-month intervals and more comprehensive follow-up assessments at annual and biennial in-person meetings. Assessments included youth self-assessments, parent assessments of youth, parent self-assessments and teacher assessments of youth. For our analysis, only baseline SM data were used.

## 2.3. Data preprocessing

The ABCD fMRI data were obtained from the National Institute of Mental Health Data Archive (NDA). We used Collection 3165 (https://nda.nih.gov/edit_collection.html?id=3165), processed by the Oregon Health & Science University Developmental Cognition And Neuroimaging Lab (DCAN, https://doi.org/10.5281/zenodo.2587210), which included preprocessed and concatenated resting-state fMRI data.[2] Preprocessing steps in the DCAN pipeline were in accordance with the HCP recommended minimal preprocessing [20]. Briefly, the T1- and T2-weighted anatomical images were used to generate subject-specific surface meshes. The functional data were corrected for motion and gradient distortions, and then co-registered to the T1-weighted image and projected into CIFTI grey ordinate space, which combines data from cortical grey matter projected onto surface mesh and subcortical grey matter in volumetric space [20]. All subsequent preprocessing and analysis steps were conducted within CIFTI space. Finally, a nuisance regression and bandpass filter (0.009–0.08 Hz) were applied to remove artefacts related to head motion and respiration [21,22]. Global signal regression was not performed. In addition to the fMRI preprocessing steps applied in Collection 3165, we further removed structured artefacts using an ICA-FIX procedure [23,24].[3] Collection 3165 also provides censor files for each subject, which denote time points that exceed 0.3 mm in head motion and only segments of 5 or more time points are preserved. To generate the final time-series files, we first removed time points flagged by the censor files and then truncated the data to include only the first 10 min to ensure that all subjects had the same number of time points. After truncating, we excluded subjects from analysis for having less than 10 min of data. This was performed in order to properly process the scans in the group-ICA pipeline and to mimic the HCP dataset in which all scans were of the same length (albeit a much longer 60 min).

It is important to note that the ABCD imaging data used in the current study were processed with the DCAN pipeline which is nearly identical to the HCP minimal processing pipeline but has an additional nuisance regression for respiratory artefacts. In addition, we elected to perform motion censoring since children generally move more than adults during resting-state fMRI scans [25]. Aside from these differences, the ABCD resting-state data were prepared in an analogous manner to the HCP resting-state data in the original study.

The SM data were obtained from the NDA ABCD RDS release (http://doi.org/10.15154/1504431). The SMs were provided in a subjects-by-SMs matrix (.Rds file), containing all subjects and their available baseline and follow-up SM data up to the ABCD 2.0.1 release. The data were already processed with qualitative variables factored, summary variables generated and missing data removed (for complete details, see https://github.com/ABCD-STUDY/analysis-nda).

### 2.3.1. Subject filtering based on imaging data

To determine which subjects were appropriate for our analysis, we applied the following data inclusion criteria, and included only subjects that met *all* of the criteria:[4]

(1) Availability of resting-state scan data and motion summary data in the NDA ABCD collection 3165.
    a. Subject must have preprocessed and concatenated resting-state fMRI data.
    b. Subject must have a motion summary file associated with the resting-state scan data, and motion data and censoring data must be present in the file.
(2) Subject must have at least 10 min of 'good' resting-state scan time (i.e. 750 remaining time points after frame removal/motion correction with a maximum frame displacement (FD) threshold of 0.3 mm).
(3) Subject's mean FD must not be anomalous in the overall distribution (i.e. it does not fall within the top 0.25% or bottom 0.25% of the motion distribution).
(4) Subject's scans must meet QC requirements, based on QC data included in the SM data matrix.
    a. At least one T1 anatomical scan which passed both QC and protocol compliance checks, for registration.
    b. At least two resting-state scans that passed both QC and protocol compliance checks.

---

[2]Rather than using the fully preprocessed derivatives distributed in Collection 3165, we elected to generate our own using the same pipeline. See §3.3 for details.

[3]We neglected to specify which ICA-FIX classifier we would use in the accepted preregistration. We therefore opted to submit the data to two analysis streams: one using an ICA-FIX classifier trained on ABCD data, which is presented in the main paper, and another using the ICA-FIX classifier trained on HCP, which was used in Smith *et al.* and for which our results are presented in electronic supplementary material, S2.3.

[4]The revised criteria used in our final analysis are presented in §3.3.

Subjects that were missing any of the aforementioned data (scan, motion or QC metrics data), had less than 10 min of 'good' scan time after motion censoring, had anomalous FD motion values, and/or did not pass the QC thresholds were removed from our sample. The cutoffs for anomalous mean FD motion values were 0.1663 mm for the top 0.25% and 0.0212 mm for the bottom 0.25%

After this initial filtering, 434 subjects were dropped due to criteria one, 1743 due to criteria two, 9 due to criteria three and 40 due to criteria four, for a total of 2226 subjects flagged for removal. As a result, 7812[5] out of the original 10 038[6] subjects remained as possible candidates for our analysis after this stage of filtering.

## 2.3.2. Subject measure filtering

Following the same procedure as Smith *et al.*, a quantitative filtering was performed in order to remove SM data unsuitable for CCA due to insufficient variance, too much missing data, or potential outliers that could skew the CCA [6,8]. Only the baseline measurements for subjects were kept (based on the earliest dated record for each subject), and all follow-up assessments were removed from consideration.

A SM was kept only if *all* of the following criteria, derived from Smith *et al.*, were met:

(1) There was enough data available.
   (a) Defined as at least 50% of subjects having data for a given SM.
(2) There was sufficient variation in the SM.
   (a) Defined as less than 95% of subjects having the same SM value.
(3) The SM did not contain an extreme outlier value based on the most extreme value from the median.
   (a) Specifically, a SM contained an extreme outlier if: $max(Y_s)$ greater than $100^*mean(Y_s)$, where $X_s$ is a vector of all subjects' values for an SM $s$, and vector $Y_s = (X_s - median(X_s))^2$.

The ABCD 2.0.1 release included a total of 64 148 SMs. After applying these criteria, 5699 SMs were dropped due to criteria one, 2765 due to criteria two and 41 383 due to criteria three, for a total of 49 847 SMs dropped, resulting in 14 301 out of the original 64 148 SMs passing as quantitatively appropriate for our analysis. For a list of all SMs and why they were dropped according to the quantitative criteria, see electronic supplementary material, file 1.

From the remaining 14 301 SMs, we then identified ABCD SMs to include in our study via a one-to-one matching of the remaining ABCD SMs with the 461 HCP SMs published by the original authors (https://www.fmrib.ox.ac.uk/datasets/HCP-CCA/). When possible, exact SM matches were selected (for example, both studies used the NIH Toolbox, and thus have identical SMs for this construct); otherwise, SMs were selected to be as similar as possible. Where multiple candidate matching SMs were present, we included all reasonable matches. This one-to-one matching identified 89 ABCD SMs which were deemed to be exact or approximate matches to HCP SMs in the original study.

In the original study, a total of nine confound variables were identified and regressed out of the SM matrix (reconstruction software version, head motion, weight, height, systolic and diastolic blood pressures, haemoglobin A1C level, brain volume and intracranial volume) [6]. However, some of these SMs were not available in the ABCD dataset or were deemed unsuitable by the quantitative exclusion; specifically, the ABCD dataset did not contain the systolic or diastolic blood pressures, haemoglobin A1C, or quarter/release (i.e. the type of reconstruction software used) data for subjects. Subject height was not included as a confound because it did not satisfy SM quantitative inclusion criteria three. In addition, we included SMs for scanner manufacturers and ABCD scan sites to account for any site- or scanner-dependent variation. Thus, the following ABCD SMs were identified as confounds to be regressed out of the SM data prior to the analysis (the ABCD SM name, as listed on the NDA, is given in parentheses):

1. Scanner site (*abcd_site*).
2. MRI scanner manufacturer (*mri_info_manufacturer*).
3. Mean frame displacement (*remaining_frame_mean_FD*).
4. Weight (*antho_weight_calc*).

[5]Actual sample size at this step was $N = 5013$. See §3.3 for details.

[6]Note that while the ABCD 2.0.1 release contains a sample size of 11 875 subjects, the DCAN Collection 3165 only contains imaging data from 10 038 subjects.

5. BMI (*anthro_bmi_calc*).
6. Cube-root of total brain volume (*smri_vol_subcort.aseg_wholebrain*).
7. Cube-root of total intracranial volume (*smri_vol_subcort.aseg_intracranialcolume*).

All 7 confound SMs were demeaned and any missing data were imputed as zero. Additional confounds were generated by demeaning and squaring confound SMs 3–7, for a total of 12 confound variables. This was done to account for any potentially nonlinear effects of these SMs [6]. These 7 confound SMs were excluded from the CCA; however, they were regressed out of the SMs that were ultimately input to the CCA. Smith *et al*. identified and excluded an additional 45 variables that were considered 'undesirable' as they were 'not sufficiently likely to be measures relating to brain function' or 'minor measures highly correlated with more major-related measures'. We matched eight of our SMs with these variables (sex, age, race, handedness, family income level, parent employment status, parent education level, parent marital status). To be consistent with the original study, they were excluded from the analysis.

After excluding the seven confounds and eight undesirable SMs, 74 SMs remained for use as the input to our analysis.[7] At this point, we removed any subjects that were missing more than 50% of the final 74 SMs ($n = 2$).[8] Thus, the final SM matrix had dimensions 7810 by 74 (subjects-by-SMs).[9] Although only this selected subset of 74 SMs was used as inputs in the CCA, we examined the following 4 SMs in our *post hoc* positive–negative axis: BMI, age, household income level and parent education level. These SMs were selected because they were analysed *post hoc* in the original study and because we deemed the ABCD measures quantitatively appropriate for correlation analysis. BMI and age are continuous variables whose magnitude has interpretable meaning. Household income level and education level were factored such that higher scores correspond to higher income or higher education level. For a list of the final 74 SMs, see appendix A; for the one-to-one ABCD/HCP SM matching, see electronic supplementary material, file 2.

### 2.3.3. Group-ICA of ABCD resting-state fMRI data

Using the preprocessed resting-state scan data, we calculated a 200-dimension group-ICA functional parcellation using FSL's MELODIC tool [26]. We used MELODIC's *temporal concatenation* option for multi-session fMRI data (https://fsl.fmrib.ox.ac.uk/fsl/fslwiki/MELODIC). After calculating the group-ICA, we used FSL's *dual regression* utility to derive the individual subject parcellated time series. Specifically, dual-regression stage-1 was used to estimate the node-time series, in which the 200-dimension group-ICA map was used as a spatial regressor against the full time-series data for each subject, estimating one parcellated time series per subject. The result is 200 nodes' time series of 750 time points (for 600 s of scan time) for each subject. Finally, we derived the subject-level functional connectivity matrices (netmats) via network modelling using the *FSLNets* toolbox (https://fsl.fmrib.ox.ac.uk/fsl/fslwiki/FSLNets). Netmats were estimated for each of the 7810 subjects[10] using partial correlation with L2 regularization ($rho = 0.01$) via FSLNets' Ridge Regression netmat option (*ridgep*). The symmetric netmats had dimensions $200 \times 200$, where each entry of the matrix was a number representing the strength of the edge between two nodes (indicated by the row and column indices). The netmat values were converted from Pearson's $r$ values to z-scores via Fisher's transformation, and a group average partial correlation network matrix was estimated by averaging the z-scored netmats across all 7810 subjects. Full correlation netmats were also calculated using FSLNets' full correlation option (corr), and a group average full correlation network was estimated by averaging the z-scored full correlation netmats across all 7810 subjects.

As in Smith *et al*., the subject-level functional connectivity matrices were combined to create a single subjects-by-edges matrix. This was accomplished by half-vectorizing each subject's symmetric functional connectivity matrix (200 dimensions) to obtain a vector of length $n(n - 1)/2$ (equal to 19 900 for $n = 200$ dimensions) per subject, and concatenating these subject-level vectors to produce the final subjects-by-edges matrix with dimensions $7810 \times 19\,900$.[11] Preparation of this matrix can be accomplished with the code provided in our project repository.

---

[7]Our revised subject filtering criteria resulted in 73 final SMs. The SM 'resiliency_6b' was removed for having too much missing data (SM inclusion criterion 1).

[8]In our revised 5013 subject sample, no subjects were *missing* more than 50% of the final SMs ($n = 0$).

[9]Due to the change in sample size (see §3.3 for details) and SM exclusion the final dimensions of the SM matrix was $5013 \times 73$.

[10]The final sample was $N = 5013$, see §3.3 for details.

[11]Due to change in sample size, the final subjects-by-edges matrix had the dimensions $5013 \times 19\,900$.

## 2.4. Data analysis

### 2.4.1. Data preparation

The CCA pipeline was based on the code provided by the original authors at their website (https://www.fmrib.ox.ac.uk/datasets/HCP-CCA/) and the steps described in the original study [6]. The CCA script was validated through an exact computational replication of the original Smith *et al.* study [10]. Aside from differing input connectome and SM data, sizes of matrices and confounds used, our analysis is identical to that of the original study. We outline the analysis steps here for clarity (shown graphically in figure 1).

The $7810 \times 74$ subjects-by-SM data matrix[12] was imported into Matlab and stored in matrix $S_1$. Matrix $S_2$ was formed from $S_1$ by using a rank-based inverse Gaussian transformation, ensuring Gaussianity for each SM. Matrix $S_3$ was then generated by regressing out the 12 confound variables (see §2.3.2) out of matrix $S_2$.

In order to run the pre-CCA PCA reduction [6,8] on the SM data, it was first necessary to account for missing data in matrix $S_3$ (1.73% of data was missing). To do so, we estimated a subjects-by-subjects covariance matrix one SM at a time for matrix $S_3$, where, for any pair of two subjects, SMs missing in either subject resulted in the comparison being ignored. The approximated covariance matrix was then projected onto the nearest valid positive-definite covariance matrix using the *nearestSPD* toolbox [27] in Matlab, resulting in matrix $S_4$ (dimensions $7810 \times 74$), which had no missing data and thus accounted for missing data without needing to impute missing SM values. There was a strong correlation ($r = 0.9999$) between the before and after covariance matrices.

$S_4$ was input to a 70-dimension PCA, generating matrix $S_5$ which was the SM data matrix input to the CCA. Although a 100-dimension PCA (as in the original study) was not possible due to having fewer valid SMs than in the original study, Smith *et al.* noted that there were no statistically significant differences in the final CCA model when the pre-CCA reduction of SMs and netmats was run with a much smaller number of PCA components (specifically, 30 instead of 100).

The functional connectomes were processed in a similar manner. First, the $7810 \times 19\,900$ subjects-by-edges data matrix[13] was imported into Matlab and stored in $N_0$. From $N_0$, two matrices ($N_1$ and $N_2$) were generated, and then horizontally concatenated to form matrix $N_3$. $N_1$ was formed by first demeaning $N_0$ column-wise, then globally variance normalizing the matrix. $N_2$ was formed by normalizing each column of $N_0$ relative to its mean, followed by dropping any columns whose mean values were too low (less than 0.1), then demeaning the matrix column-wise and finally globally variance normalizing it. The 12 confound variables were then regressed out of $N_3$, forming matrix $N_4$. $N_4$ served as the input to the 70-dimension PCA to reduce dimensionality, estimating the top 70 eigenvectors of the connectomes to form matrix $N_5$, which was the connectome data matrix input to the CCA.

### 2.4.2. Canonical correlation analysis

The CCA was run using the Matlab *canoncorr* function (from the Statistics and Machine Learning Toolbox), using the 70-eigenvector matrices for the SMs ($S_5$) and connectomes ($N_5$) as inputs. The CCA estimated 70 modes, optimizing de-mixing matrices $A$ and $B$ such that the resulting subject-weight matrices $U = N_5{}^*A$ and $V = S_5{}^*B$ (both had dimensions $7810 \times 70$[14]) were maximally similar to each other. We then determined the correlation between the subject connectome and SM weights for each of the 70 CCA modes by correlating the corresponding column pairs in matrices $U$ and $V$. The Pearson's $r$ value for each CCA mode indicated the strength for which a mode of covariation exists in both the subject connectomes and SMs.

To estimate the significance of the correlation between the weights of each CCA mode (i.e. correlation between the corresponding columns of matrices $U$ and $V$), we performed a 100 000 permutation-based significance test in which the CCA was re-run on each permutation of the SM matrix to generate a null distribution of CCA results. As in the original study, the permutations (preserving family structure) were generated using the package *hcp2blocks* (https://github.com/andersonwinkler/HCP/blob/master/share/hcp2blocks.m) [28].[15] Sibships were determined based on subject zygosity and their parent/family IDs; all related subjects were matched according to their family IDs. Within these family

---

[12]Due to a change in sample size and SM exclusion, the dimensions of $S_1$ were $5013 \times 73$.

[13]Due to a change in sample size, the dimensions of $N_0$ were $5013 \times 19\,900$.

[14]Due to a change in sample size, the final dimensions of $N_5$ and $S_5$ were $5013 \times 70$.

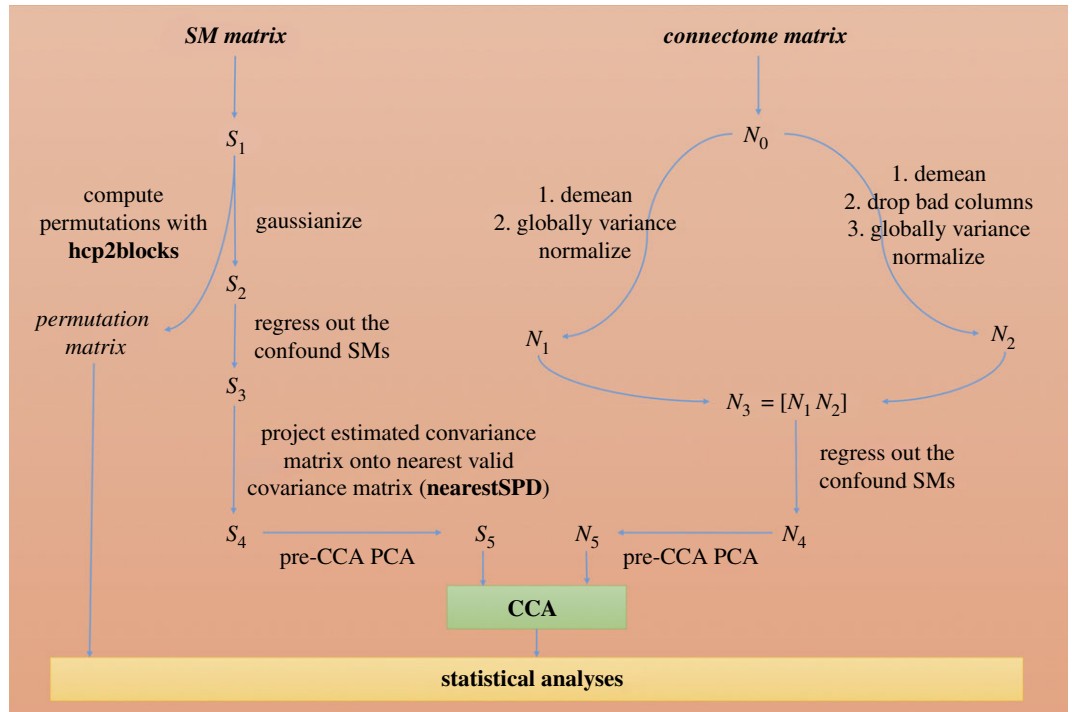

**Figure 1.** Outline of data analysis, highlighting the pre-processing stages for subject measures (left-hand side) and connectomes (right-hand side) immediately prior to the CCA. $N_0$–$N_5$ and $S_1$–$S_5$ are the names of variables in our Matlab code.

groupings, MZ twins were permuted, DZ twins were permuted, and non-twin siblings were permuted. After the subject-level permutations were created, entire families that contained the same types of sibships and number of children were permuted. As discussed in §2.1, to match the procedures of the original study, any twins/triplets whose zygosity data were unavailable were treated as non-twin siblings when generating permutations. Since there are many related subjects in the ABCD dataset (1339 subjects were twins, 1245 were non-twin siblings) the family structure was maintained when permuting so that a valid null distribution could be built.

The null distributions were used to calculate thresholds for significance in the percent variance explained in the connectomes and SMs by each CCA mode. Specifically, the 5th and 95th percentiles of variance in the null distributions were used to determine which of the first 20 CCA modes explained a significant percentage of the total variance in the original SM ($S_2$) and connectome ($N_0$) data matrices. Connectomes and SMs were analysed separately.

### 2.4.3. Post hoc analyses

To understand which SMs were most strongly correlated with the primary mode and determine the directionality of the relationship (a positive or negative correlation), we generated a positive–negative axis that quantified the level of variance explained by each SM. This analysis included the final 74 SMs that were input to the pre-CCA reduction, and 4 additional SMs (BMI, age, household income level, parent education level; see appendix A) for a total of 78 SMs whose raw data were stored in a new matrix, $S_7$ (dimensions 7810 × 78 subjects-by-SMs).[16] Matrix $S_8$ was then calculated by regressing the 12 confound variables (identified in §2.4.1) out of matrix $S_7$ (similar to how matrix $S_3$ was originally calculated). We correlated each column of matrix $S_8$ with the primary mode SM weights (i.e. the first column of matrix $V$) to find Pearson's $r$ values for each of the 78 SMs against the primary mode, and then plotted these correlations on an axis. In addition to the Pearson's $r$ correlation values, we calculated z-score for each SM and the percentage of variance in the primary

---

[15]Instead of using *hcp2blocks*, we used a modified version of this script to generate our permutation blocks. The code was modified to handle data from multiple study sites, and can be found at https://github.com/andersonwinkler/ABCD/blob/master/share/abcd2blocks.m.

[16]Due to a change in sample size and SM exclusion, the dimensions of $S_7$ were 5013 × 77.

CCA mode that was explained by each SM. The percent variance was calculated by solving the following expression for each SM:

$$\text{Variance}_{SM_i} = \frac{\text{variance}(V_1 * pinv(V_1) * S_{8_i})}{\text{variance}(S_{8_i})},$$

where $pinv()$ is Matlab's pseudo-inverse function, vector $V_1$ is a demeaned copy of the primary CCA mode weights for SMs (a $7810 \times 1$ vector)[17] and vector $S_{8_i}$ (also a $7810 \times 1$ vector)[18] is a demeaned copy of the raw values for the $i$th SM in matrix $S_8$.

To facilitate interpretation of our ICA and connectivity results, we submitted the nodes derived from the group-ICA to a hierarchical clustering algorithm. As in Smith *et al.*, we entered the group-averaged full correlation network (all 200 nodes) into Ward's clustering algorithm implemented in Matlab. The low-dimensional results were then examined and assessed for correspondence with large-scale clusters observed in Smith *et al.* (e.g. sensory, motor, dorsal attention and DMNs). To calculate the correlation between the CCA connectome-modulation weights (from the initial CCA) and the original population mean connectome, we first obtained the relative weights of involvement of the original set of connectome edges by correlating the primary CCA mode connectome weight vector $U_1$ (i.e. the first column of matrix $U$) against $N_0$, resulting in the 'full length' edge weight vector $A_{F1}$ (a 19 900 element vector), effectively mapping the primary CCA mode onto the original connectome data matrix. $A_{F1}$ was then correlated against the original population mean connectome.

In addition to the 100 000 permutation-based significance test of the main CCA result, a 80–20 split train-test cross-validation was performed in which approximately 80% (approx. 6249) of subjects were used in a 'training' CCA, and the remaining 20% (approx. 1562) of subjects were left out as a test validation set.[19] Since the ABCD dataset contains sibships, the training and test subsets were generated without splitting any families across the subsets. To calculate the 'training' CCA, we applied the same steps outlined in §2.4.1 to the training subset of the connectome and SM data to calculate the de-mixing matrices $A_{\text{train}}$ and $B_{\text{train}}$ such that $U_{\text{train}} = N_{5,\text{train}} {}^* A_{\text{train}}$ and $V_{\text{train}} = S_{5,\text{train}} {}^* B_{\text{train}}$.

We then evaluated the primary mode of the 'training' CCA to determine how similar it was to the initial CCA result (the first CCA calculated with all 7810[20] subjects; see §2.4.2). The primary mode weight vectors for connectomes and SMs ($U_{\text{train},1}$ and $V_{\text{train},1}$, respectively) were correlated against the primary mode weight vectors for connectomes and SMs from the original analysis (vectors $U_1$ and $V_1$).

To test the performance of the 'training' CCA, the CCA connectome and SM de-mixing matrices, $A_{\text{train}}$ and $B_{\text{train}}$, from the train dataset were multiplied into the left-out SM and connectome matrices in order to estimate subject weight matrices $U_{\text{test}}$ and $V_{\text{test}}$ for the test dataset. The subject weight matrices were estimated as follows:

$$U_{\text{test}} \approx N_{5,\text{test}} * A_{\text{train}}$$

and

$$V_{\text{test}} \approx S_{5,\text{test}} * B_{\text{train}}.$$

The matrices $N_{5,\text{test}}$ and $S_{5,\text{test}}$ were calculated from the testing subset of the connectome and SM data via steps similar to those outlined in §2.4.1. The connectome and SM weight vectors for the primary mode of the 'testing' CCA (i.e. the first column of $U_{\text{test}}$ and $V_{\text{test}}$, denoted $U_{\text{test},1}$ and $V_{\text{test},1}$) were then correlated. In order to measure the significance of correlation between the 'testing' CCA primary mode connectome and SM weight vectors $U_{\text{test},1}$ and $V_{\text{test},1}$, we performed a 1000-permutation test (respecting family structure). This test-train and significance testing process was repeated 10 times, each time with a randomly selected subset of test and train subjects, and the average correlation of $U_{\text{test},1}$ and $V_{\text{test},1}$ from all 10 runs was calculated.

---

[17]Due to a change in sample size, the dimensions of the vector were $5013 \times 1$.

[18]Due to a change in sample size, the dimensions of the vector were $5013 \times 1$.

[19]Due to a change in sample size, 80% of the sample was $N = {\sim}4010$ and 20% of the sample was $N = {\sim}1003$.

[20]Due to a change in sample size, the first CCA was calculated with 5013 subjects.

[21]After filtering based on our revised criteria, the final figures were as follows: 5013 subjects (47.9% male, 52.1% female). Age in years (mean, standard deviation, range): 9.99, 0.625, 9.0–10.21. Race: 59.6% White, 10.4% Black, 17.5% Hispanic, 1.9% Asian, 10.5% Other. Sibships: 360 MZ, 514 DZ, 810 non-twin siblings (687 siblings+111 twins missing zygosity+12 triplets), 3329 single. See §3.3 for more information.

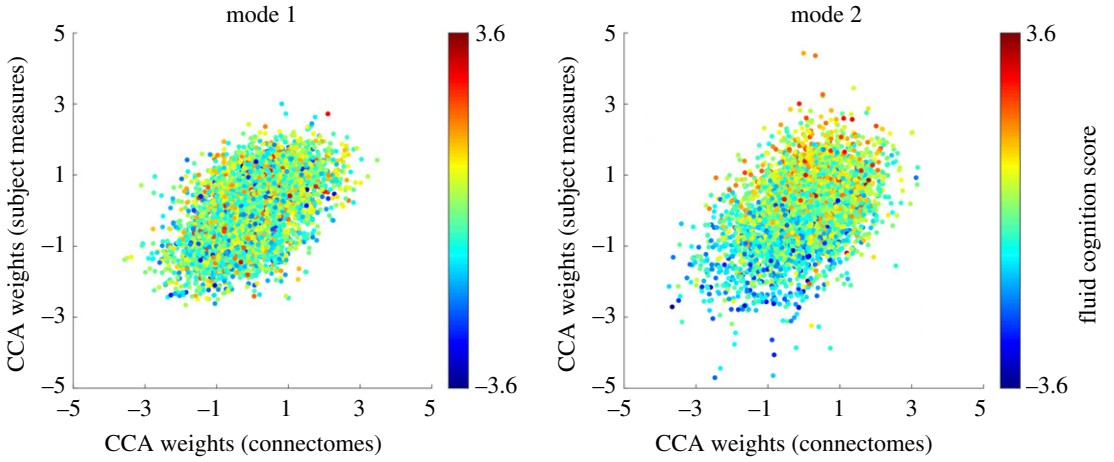

**Figure 2.** Correlation between SM (subject measures) and connectome weights for CCA Mode 1 ($r = 0.53$, permutation $p < 10^{-5}$) and Mode 2 ($r = 0.44$, $p < 10^{-5}$). Each dot represents one subject ($n = 5013$). As an example SM, points are coloured by subjects' fluid cognition score as measured by the NIH Toolbox.

## 2.5 Code availability

The code to run this entire pipeline is available on this project's Github repository (https://github.com/nih-fmrif/abcd_cca_replication). We have also uploaded a compressed archive of the code in the Github repository to the OSF project (http://doi.org/10.17605/OSF.IO/HMC35) which also contains the full text of the pre-registration. Questions about the code may be posted as issues on the github project or emailed to the corresponding author directly.

# 3. Results

## 3.1. Canonical correlation analysis

The CCA yielded four modes whose correlation between the SMs and connectomes weights were statistically significant by permutation test (Mode 1: $r = 0.53$, $p < 10^{-5}$; Mode 2: $r = 0.44$, $p < 10^{-5}$; Mode 3: $r = 0.26$, $p < 10^{-4}$; Mode 4: $r = 0.25$, $p < 0.01$, corrected for multiple comparisons of all modes). Scatterplots in figure 2 illustrate the relationship between SM and connectome weights in CCA Modes 1 and 2 (see electronic supplementary material, figure S1, for Modes 3 and 4). Following the approach of Smith *et al.*, we colour the points by an example SM; here we chose fluid cognition as measured by the NIH Toolbox as it was most similar to the fluid intelligence measure used in fig. 1*b* from Smith *et al.* While fluid cognition does not appear to be associated with the relationship between SM and connectome weights in Mode 1, we observed a positive relationship in Mode 2 such that subjects with greater Mode 2 weights for both SMs and connectomes have higher fluid cognition scores. However, the relationship between CCA weights and fluid cognition in figure 2 is meant to be qualitative. In our *post hoc* positive–negative axis analysis we quantitatively relate all SMs to the primary CCA mode.

We next examined the variance from the full SM and connectome matrices explained by the CCA modes by converting the Pearson's $r$ correlation values between the CCA weights and the demeaned SM and connectome matrices to $R^2$. We then compared the observed $R^2$ values to the null distribution of $R^2$ values obtained from our permutation test in which we shuffled the SM matrix 100 000 times. z-scores for the observed $R^2$ values were calculated by subtracting the observed $R^2$ from the mean of the null $R^2$ distribution divided by the null standard deviation of $R^2$ values. For the connectome matrix, Modes 1 and 2 explained a significant amount of variance compared to the permutation-test derived null distribution (Mode 1: 0.096% variance explained, $Z = 2.99$; Mode 2: 0.097% variance explained, $Z = 3.16$). For the SM matrix, Modes 2 and 3 explained a significant amount of variance (Mode 2: 5.58%, $Z = 8.56$; Mode 3: 3.23%, $Z = 3.64$). Figure 3*a,b* shows the connectome variance and SM variance explained for each of the top twenty modes.

From these results, we determine that Mode 2 is the primary, 'strongest' mode according to our replication criteria (see §1.3.1), since it maximally explains variance in both SM and connectome matrices (criterion 1) and exhibits a statistically significant correlation between SM and connectome

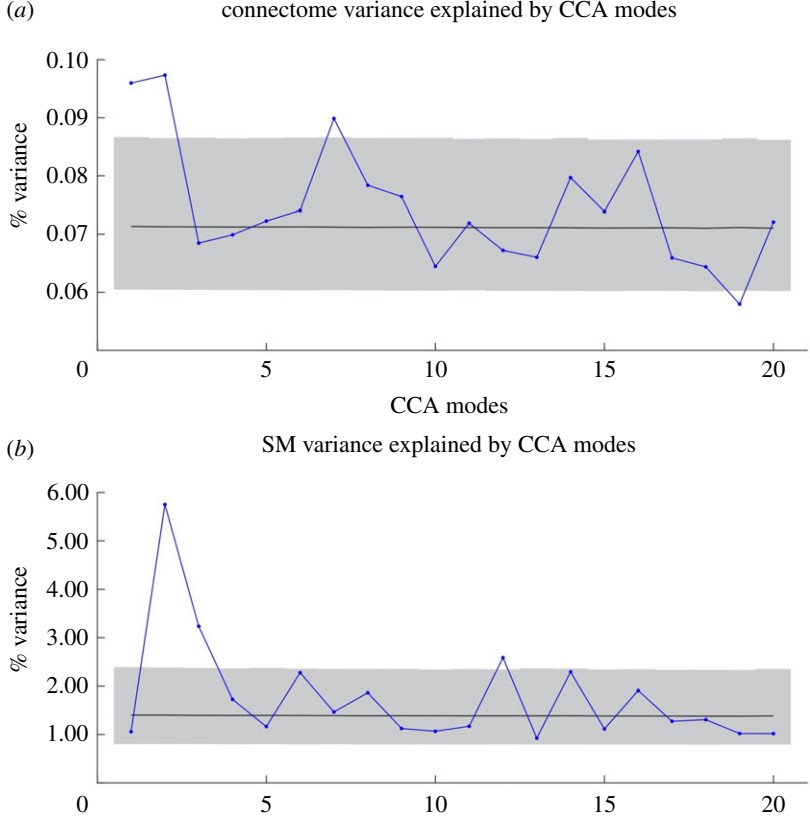

(a) connectome variance explained by CCA modes

(b) SM variance explained by CCA modes

**Figure 3.** Variance explained in the original (a) connectome and (b) SM matrices. Blue dots and lines correspond to the observed variance explained. The mean variance explained of the null distribution is in black and the 5th and 95th percentiles of the null distribution are in grey.

weights (criterion 3). The z-score for SM variance explained by Mode 2 was greater than the next highest z-score by more than a factor of 2 (criterion 2).

The only replication criterion not met is criterion 2, which addressed the variance explained by the connectome and SM weights. We expected that the connectome variance explained by the primary CCA mode would be a factor of 2 greater than the next highest mode. Instead, we observed that Modes 1 and 2 explained similar amounts of variance in the connectomes (Mode 1: $Z = 2.99$, Mode 2: $Z = 3.16$). Although not part of our criteria, we did note that both of these z-scores were a factor of 2 larger than Mode 3 ($Z = -0.34$). With regard to the SM matrix, we expected the primary mode's variance explained z-score to be a factor of 3 larger than the next highest mode. Instead, we observed that the z-score of Mode 2 (8.56) was a factor of 2.35 greater than the next highest mode (Mode 3: $Z = 3.64$). Thus while our results are numerically similar to our expectations, the magnitude of variance explained by the primary CCA mode did not reach our *a priori* hypotheses.

To validate the CCA results, which were calculated on the full SM and connectome matrices, we then submitted the data to a 80–20 train-test split cross-validation procedure. Here we implemented a CCA on 80% of the sample and then estimated the subject weight vectors for Mode 2 in the left-out 20%. The weight vectors were then correlated and compared against 1000 permutations where we shuffled the SM matrix and performed the same analysis. This procedure was repeated 10 times. The mean correlation between $U_{\text{test}}$ and $V_{\text{test}}$ was $0.22 \pm 0.03$, whereas the mean correlation for the permutation was 0.002. In all 10 iterations, the cross-validated correlations were significantly different from the null-permuted correlations ($p < 0.001$), thus confirming our main CCA results. Further, our 80–20 train-test split results are numerically similar to the results from Smith *et al.* (i.e. mean correlation between $U_{\text{test}}$ and $V_{\text{test}}$ in the original paper was 0.25). Notably, while still statistically significant, the mean correlation between the $U_{\text{test}}$ and $V_{\text{test}}$ ($r = 0.22$) is smaller than the corresponding correlation from the full CCA ($r = 0.44$). We attribute this decrease in correlation to the possibility that the CCA model defined on the full dataset may be overfit and a generalizable correlation between canonical variates would likely be smaller in magnitude.

The results presented thus far used data from a preprocessing pipeline that removed structured artefacts using an ABCD-trained ICA-FIX classifier. However, to remain as close to the original study as possible, we also repeated the main CCA and variance analysis on data preprocessed with an ICA-FIX classifier trained using data from the HCP. While there were small numeric differences in the results from these two pipelines, the overall findings are consistent, suggesting that our results are not dependent on which ICA-FIX classifier we use. See electronic supplementary material, S2.3.

## 3.2. *Post hoc* analyses

We next sought to determine the relationship between the CCA modes and specific SMs and connectome edges. Here we focus on Mode 2 as this is the primary mode according to our replication criteria (see §1.3.1).

As in Smith *et al.*, we found a positive–negative axis where positively valenced SMs (e.g. measures of neurocognition, memory, executive function, parental education) correlated positively with the CCA mode, whereas negatively valenced SMs (e.g. Child Behaviour Checklist ADHD, rule breaking behaviour, conduct disorder) correlated negatively with the CCA mode. Interestingly, no negatively valenced SMs were on the positive end of the spectrum and no positively valenced SMs were found on the negative end. See figure 4 for a depiction of the ABCD positive–negative axis for Mode 2 and electronic supplementary material, figure S2, for an unthresholded positive–negative axis that includes all SMs. Further, we also find consistent positive–negative axis results in the HCP-trained ICA-FIX pipeline. See electronic supplementary material, figure S5, for the axis thresholded at $r = \pm 0.2$ and electronic supplementary material, figure S6, for the unthresholded axis.

Finally, we examined the relationship between specific connectome edges and the primary CCA mode (Mode 2). Here we calculated correlations between subject-specific connectomes and the connectome weights from the primary CCA mode. Figure 5 shows the top 30 edges most strongly related to the primary mode, with node clusters derived from clustering the full set of connectomes using Ward's clustering algorithm. In figure 5 we observe that a variety of nodes are represented within the strongest edges, including sensory regions as well as higher-order association cortices topographically similar to the posterior midline and temporo-parietal regions of the DMN. Interestingly, compared to Smith *et al.* we do not observe a clear organization in which all nodes topographically similar to the DMN (e.g. nodes 2, 5, 8, 10, 11, 14, 20, 23, 26 in figure 5) are clustered together. Instead, these nodes are spread out among different clusters; however, as in Smith *et al.*, the connectome edges between these nodes are all positively related to the primary CCA mode. See electronic supplementary material, figure S7, for the top 30 strongest edges associated with the primary mode in the HCP-trained ICA-FIX pipeline.

## 3.3. Deviations from protocol

On closer inspection of the ABCD dataset, we realized that deviations from our pre-registration were necessary to more closely adhere to the analysis methods used in Smith *et al*. All of the deviations detailed below were made while preprocessing the data. The final CCA mode calculation, on which successful replication was defined, was not conducted prior to the deviations and did not influence our decisions. The accepted Stage 1 protocol is available on the project's Open Science Foundation page (http://doi.org/10.17605/OSF.IO/HMC35).

In our pre-registration, we planned to use the fully preprocessed and concatenated ABCD resting-state data distributed by DCAN. The DCAN pipeline removes motion and respiratory artefacts through nuisance regression and filtration [21], whereas Smith *et al*. removed structured artefacts using ICA + FIX [23,24]. If we were to run ICA + FIX on the already 'cleaned' preprocessed DCAN pipeline data, this would be a significant departure from the methods of Smith *et al*. We, therefore, decided to use the DCAN's BIDS-formatted input data and their abcd-hcp-pipeline (https://github.com/DCAN-Labs/abcd-hcp-pipeline, v. 0.0.1) to do preprocessing in accordance with the HCP recommended minimal processing (the first stages of the DCAN pipeline are a wrapper for the HCP minimal pipeline, which was used in Smith *et al*.).

In our stage 1 pre-registered manuscript, we failed to specify which ICA-FIX classifier we would use to classify and remove structured artefacts. Therefore we elected to run two versions of our pipeline: one using an ICA-FIX classifier that we trained using ABCD data and another using the ICA-FIX classifier distributed with the ICA-FIX software which was trained using HCP data. We present the results from the ABCD-trained ICA-FIX pipeline in the main paper and the HCP-trained ICA-FIX pipeline in

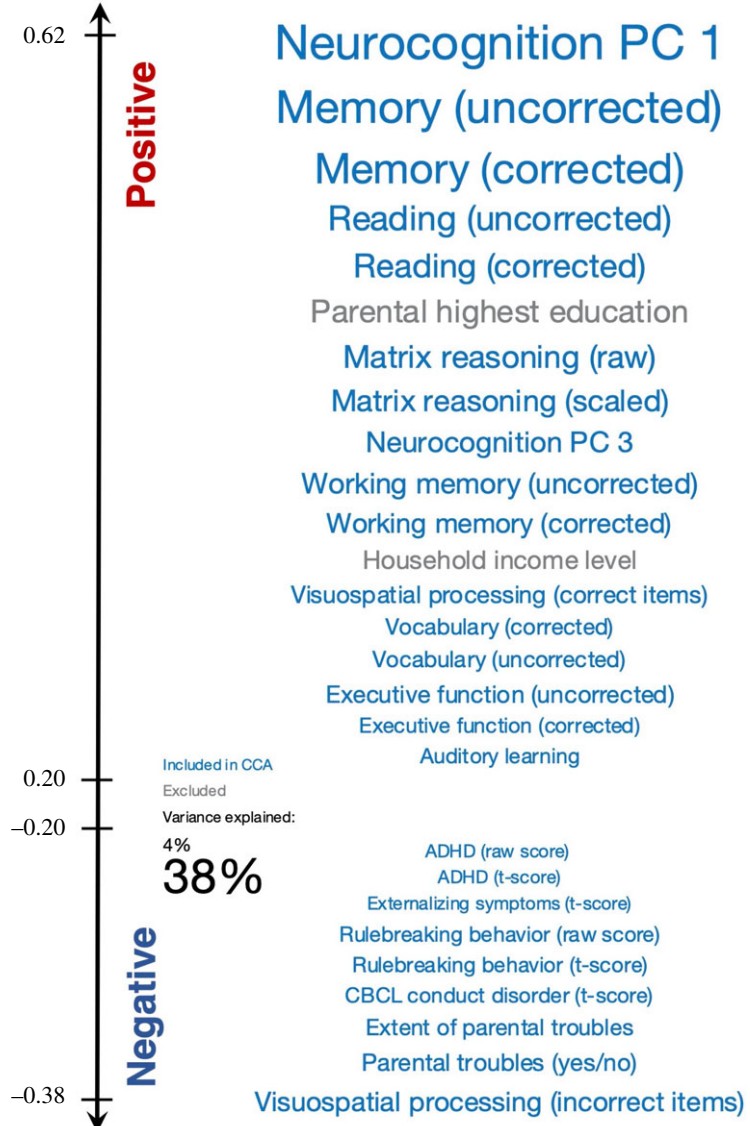

**Figure 4.** The SMs most strongly related to the CCA Mode 2 are organized by correlation value. SMs were thresholded at $r = \pm 0.2$. SM label font size corresponds to the amount of SM variance explained by the CCA mode (smallest font = 4%, largest font = 38%). Blue font colour indicates that the SM was included in the CCA; grey indicates that the SM was not included.

S2.3 of the electronic supplementary material. See S1.1 in the electronic supplementary material for information regarding ICA-FIX classifier signal and noise component classification performance and methods regarding training our own classifier.

Since we elected to not use the preprocessed, concatenated fMRI time series included in the DCAN ABCD release, it was necessary to examine the quality of the four individual fMRI runs for each subject before passing them to ICA-FIX. Here we noticed many runs had low numbers of time points (i.e. less than the full acquisition protocol) and/or high mean framewise displacement (FD) values, which could lead to spurious results. Thus, we applied the following revised data inclusion criteria. The addition of the following inclusion criterion 1 is the only deviation in inclusion proctol. Inclusion criteria 2 and 3 remained the same as the pre-registered protocol.

1. Subject must have at least two 'good' runs of resting-state data (see below).
2. Subject must have a total of at least 10 min (600 s, 750 time points) of post-censoring resting-state data (see below for time point censoring criteria).
3. Subject must have at least one T1-weighted image that passes QC and protocol compliance metrics provided by ABCD.

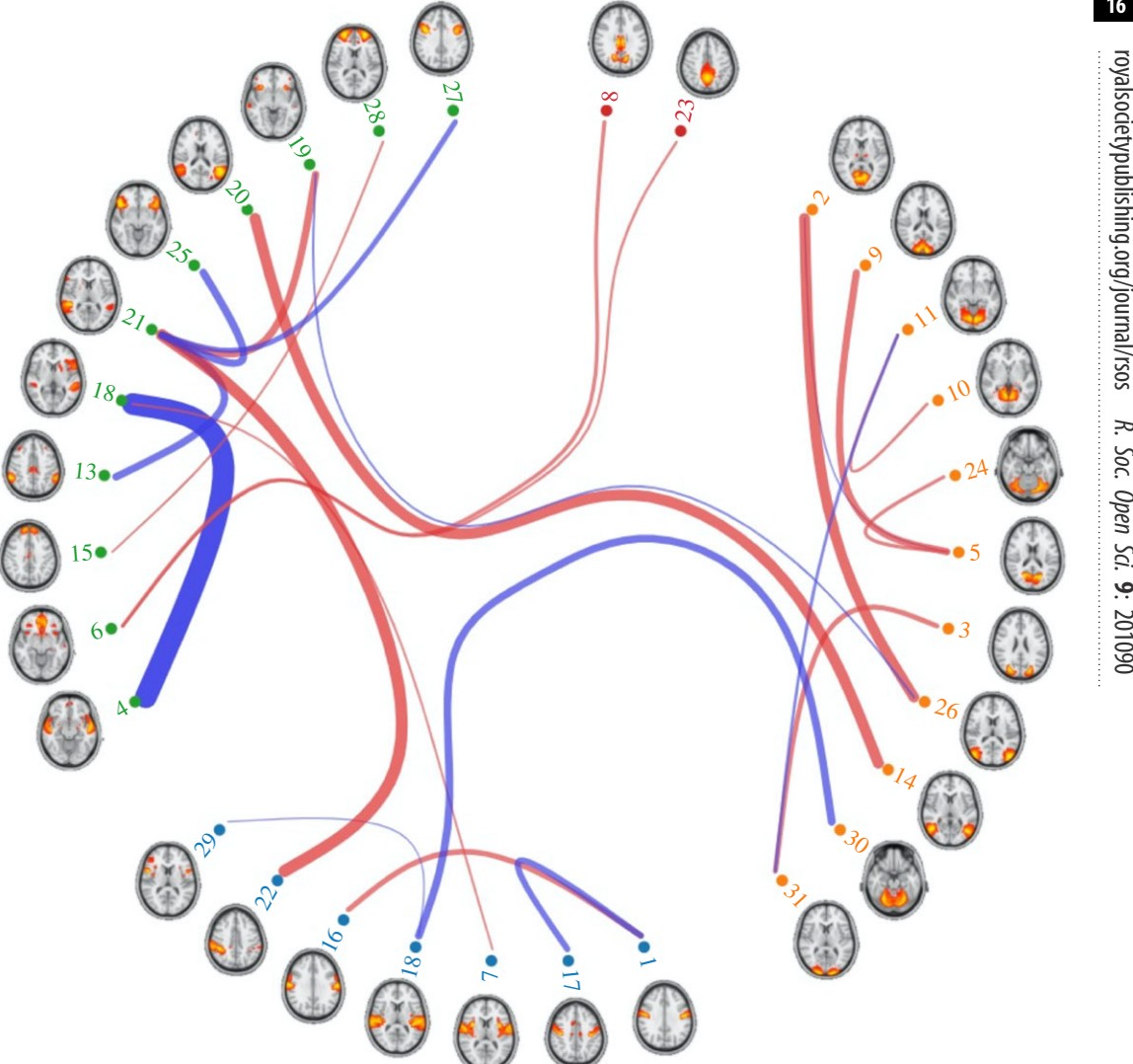

**Figure 5.** The 30 connectome edges most strongly correlated with CCA Mode 2. Nodes are organized according to a hierarchical clustering algorithm. Links are coloured according to the sign of the correlation between population mean functional connectivity and CCA mode. Thickness of the links depict the strength of the correlation between functional connectivity and CCA mode.

Using published studies that use the ABCD fMRI data as a guide [11,29,30], a resting-state run was deemed 'good' if it met all of the following criteria:

1. Mean FD for the run is less than 0.3 mm.
2. The pre-censoring run length (i.e. length of the raw scan) is at least 50% of the expected length of 380 time points (i.e. it must be at least 190 time points, or approximately 150 s long).

Runs that passed the above criteria were then submitted to ICA-FIX to remove structured artefacts. Censor files were then created using an FD threshold of 0.3 mm. To create the censor files, we used head motion parameters generated during pre-processing to flag time points that exceeded an FD of 0.3 mm for removal. In addition, segments of the time series were also flagged for removal if fewer than 5 consecutive time points had an FD displacement below the threshold. After ICA-FIX, runs were censored (i.e. time points flagged for removal were removed from the time series) and concatenated. Finally, we truncated the time series to 10 min for all subjects.

After the 10 038 subjects in Collection 3165 were filtered according to the new criteria and only those with 'good' time points were retained, 5013 subjects remained as possible candidates for our analysis.

This sample is smaller than the 7810 described in our pre-registered protocol because our new analysis excluded individual runs that lacked sufficient time points or exceeded a mean FD of 0.3 mm.

The SM filtering protocol was unchanged. However, given the revised sample of 5013 subjects, SM 'resiliency_6b' was removed for having too much missing data (i.e. failed to meet SM inclusion criterion 1; see §2.3.2), leaving 73 final SMs. No subjects in the new 5013 subject sample were dropped due to missing 50% or more SM data.

Our final sample ($N = 5013$) had the following demographic breakdown: 47.9% male, 52.1% female; age in years (mean, standard deviation, range): 9.99, 0.625, 9.0–10.21; race: 59.6% White, 10.4% Black, 17.5% Hispanic, 1.9% Asian, 10.5% Other; sibships: 360 MZ, 514 DZ, 810 non-twin siblings (687 siblings + 111 twins missing zygosity + 12 triplets), 3329 single.

# 4. Discussion

In this study we have shown that a single axis of co-variation spanning 'positive' and 'negative' attributes links diverse participant characteristics with specific patterns of brain connectivity in 9- and 10-year-old children. We set out to replicate the landmark paper from Smith *et al.*, which used the HCP dataset to examine the multidimensional relationship between functional brain network organization at rest and a variety of SMs, including metrics of cognitive function and lifestyle. Using CCA, the authors found a primary mode of correlation that explained a statistically significant amount of variance in both the functional connectomes and SMs. Further, when correlating the primary mode with specific SMs, the authors found that these correlations were organized along a positive–negative axis, such that positively valenced SMs (e.g. fluid intelligence) exhibited a positive correlation with the primary CCA mode, and the negatively valenced SMs (e.g. anger) exhibited a negative correlation. The authors also found that connectivity within the DMN was positively correlated with the primary CCA mode.

Our analysis met 2 of the 3 strict numerical replication criteria described in our pre-registration (see §1.3.1). For our first criterion, we found that Mode 2 of our CCA explained a significant amount of variance in both the connectome and SM matrices, according to a permutation test. For our third criterion, the connectome and SM CCA weights for Mode 2 exhibited a statistically significant positive correlation. Only our second criterion was unsuccessful. We expected that the primary CCA mode would explain a majority of the variance relative to the other modes, as indicated by z-scores of a factor of 2 for the connectome matrix and a factor of 3 for the SM matrix greater than the next largest z-score. In hindsight, we would judge the specific numeric magnitude factor criteria as overly conservative and not necessarily a critical finding of the original study. Further, one noteworthy difference between our findings and Smith *et al.* is that the primary mode in the original paper was Mode 1, whereas we observed Mode 2 as primary. CCA modes are ordered according to the magnitude of correlations between the corresponding linear low-rank projections of the left and right input matrices—that is, mode order relies on the strength of the relationship between the canonical variates [8]. While our criterion 3 for a successful replication dealt with the statistical significance of the correlation between these canonical variates, our criteria 1 and 2 focused on the variance that the CCA modes explained in the original SM and connectome matrices separately. Thus, while our observed Mode 1, by definition, explained the most variance in the *relationship* between SM and connectome matrices, Mode 2 was more successful at explaining the variance in each matrix *separately*. As a result, Mode 2 met our criteria for the primary mode. Further, while Smith *et al.* found a single statistically significant mode, we found four, though only one met our criteria as primary. Subsequent work examining the multidimensional relationship between brain imaging and SMs suggests that multiple interpretable modes may exist and account for a variety of brain-behaviour dimensions [31,32]. Despite the difference in which mode was primary, in our view our analysis shows that 9- to 10-year-old children exhibit the same basic phenomenon that was reported by Smith *et al.* in young adults: a surprising amount of variation in brain connectivity can be explained by a single axis of 'positive' and 'negative' attributes.

Further, our analysis has shown that this brain-behaviour relationship is robust to significant heterogeneity in acquisition and analyses. First, the HCP dataset used in the original study was collected at a single site on a single scanner [7], whereas the ABCD dataset was collected across 21 sites throughout the USA and multiple scanner manufacturers and models [15]. Second, we have shown that our findings remain consistent through two preprocessing pipelines: one which removed structured artefacts using an ICA-FIX classifier trained on ABCD data and another which leveraged the ICA-FIX classifier trained on the HCP dataset, which was used in the original study. With such

heterogeneity in the ABCD dataset relative to the HCP dataset, it is surprising that the results from Smith *et al.* replicate so well, since previous work has shown poor reliability when combining data from multiple sites using methods such as intraclass correlation coefficient [33]. While we did observe a lower correlation between SM and connectome weights compared to Smith *et al.* (original study: $r = 0.87$; current study: $r = 0.44$, for the primary mode), it is possible that the current analysis is not as susceptible to heterogeneity compared to other methods [33] since the data undergo both deconfounding and dimensionality reduction prior to the CCA (which is itself another form of dimensionality reduction between two sets of data [8]). The conceptual replication of a finding that originated using a relatively homogeneous dataset such as HCP within the more heterogeneous ABCD dataset offers hope in the face of the so-called replication crisis in psychological science more broadly [34].

The findings from Smith *et al.* are also robust to the many differences in demographics between the subjects of HCP versus those of ABCD. The ABCD dataset attempts to reflect the race and ethnicity breakdown of the United States in general [35], whereas the HCP dataset is comparatively more homogeneous [7]. Moreover, the current study extends the results from the young adults (ages 22 to 35 years) of HCP into the 9- and 10-year-old baseline sample of ABCD. Thus, we find that a primary mode of correlation between functional connectomes and SMs generalizes across multiple sources of variability in data acquisition and subject demographics, suggesting that this high-dimensional relationship may in fact be characteristic of general human cognition.

As in the original study, we also found evidence for a positive–negative axis linking specific SMs to the primary CCA mode. Here SMs typically regarded as positive qualities (e.g. working memory, executive function, parental income) were positively correlated with the primary mode, whereas SMs typically regarded as negative qualities (e.g. conduct disorder, ADHD symptoms, rule-breaking behaviour) were negatively correlated with the primary mode. Smith *et al.* noted that the primary mode could be evidence for a neural correlate of the general intelligence $g$ factor [36]. Interestingly, SMs on the positive end of the axis included both measures of neurocognition as well as two parental measures (highest level of education and income) that have been identified by the World Health Organization as key factors in the Social Determinants of Health [37]. They also serve as proxy measures of socioeconomic status (SES) which has been previously shown to have a strong relationship with brain function [38,39]. SES is a multidimensional construct [39] and many of the relationships between SES and brain function are nonlinear and/ or moderated by other factors [40,41], which warrants some caution in the interpretation of the current study. However, the fact that these measures were among the strongest correlations with the dominant mode demonstrates a need for expanding variables related to the social determinants of health in future studies to complement and contextualize standard neurocognitive and clinical measures.

On the negative-correlation side of the positive–negative axis, the appearance of clinical measures such as subscales of the Child Behaviour Checklist [42] for ADHD, conduct disorder, rule-breaking behaviors and externalizing symptoms are of particular interest since many symptoms of behaviour disorders emerge during adolescence [43]. Future work should determine the predictive accuracy of this axis. For example, the ability to predict behaviour disorder symptoms from functional connectivity (and vice versa) could be of great utility to clinicians. In addition, future work should extend this analysis into younger children to examine a possible dynamic emergence and clinically predictive accuracy of the positive–negative axis.

Finally, we also found a specific pattern of functional connectivity edges that was related to the primary CCA mode. The nodes whose edges were most strongly correlated with the CCA included nodes topologically similar to sensory areas and the DMN. As in Smith *et al.*, we found that connectivity within the DMN was positively correlated with the primary CCA mode. The DMN is a network of distributed brain regions implicated in introspective and abstract thought, social cognition and autobiographical memory [44]. However, while Smith *et al.* found nodes representing a broad swath of the DMN within the edges most strongly related to the CCA mode, we only found that edges between nodes within the posterior midline and temporo-parietal areas were most strongly related to the CCA mode. Further, while the default mode nodes in Smith *et al.* all clustered together, we observed that these nodes were represented in multiple clusters.

This is consistent with the findings on the development of the DMN and higher-order association cortices in general, such that these regions undergo a protracted development into early adulthood [45]. That is, with age the DMN shifts from a weakly connected set of nodes to a cohesive network,

which could relate to the development of increasingly complex and abstract cognitive abilities [46]. We speculate that the aforementioned SES measures are relevant to the instantiation and subsequent development of the brain's DMN and that the positive–negative axis strengthens throughout development, which could serve as a biomarker for at-risk children and adolescents. However, since we only observe that nodes within a subset of the DMN are most strongly related to the primary mode and that the baseline ABCD sample is a relatively narrow age range (i.e. 9- to 10-year-olds), it is possible that we are only tapping into one slice of a dynamic developmental process. Future work should leverage the upcoming longitudinal releases within ABCD to examine the relationship between functional connectivity edges and a primary CCA mode across adolescence, as well as the relationship between SMs and the development of the DMN.

In conclusion, the current study sought to replicate Smith *et al.* and extends their findings into the more heterogeneous ABCD dataset. We found a primary mode of correlation between brain functional connectivity and SMs meeting two of three pre-registered numerical criteria. We also found evidence of the positive–negative axis first reported by Smith *et al.* where positive SMs were positively correlated with the primary mode and negative measures were negatively correlated. Finally, like Smith *et al.*, we also found that connectivity within regions of the DMN were positively correlated with the primary mode, although this pattern of results in the ABCD dataset was not as dominated by default mode, and connections involving several other brain regions emerged as significantly linked to the primary mode; this may reflect developmental differences. The current study is situated within more recent efforts to examine the multidimensional relationship between multiple imaging derived phenotypes, physiological measures, genomics and behaviour showing interpretable modes related to phenotypes such as fluid intelligence, handedness and cardiovascular disease [29,30]. The ABCD dataset contains a similar diverse array of measures spanning these different modalities. Here we have demonstrated the feasibility of these ICA-based methods with this unique developmental dataset. Given the robustness of the current replication, we expect future studies (using more modalities and more recent approaches to data fusion) will show both replication of previous results (e.g. handedness) and discover new, interpretable modes of variation in brain structure and function related to child and adolescent development.

Data accessibility. ABCD Data. Data used in the preparation of this article were obtained from the Adolescent Brain Cognitive Development (ABCD) Study (https://abcdstudy.org), held in the NIMH Data Archive (NDA). This is a multi-site, longitudinal study designed to recruit more than 10 000 children age 9–10 and follow them over 10 years into early adulthood. The ABCD Study is supported by the National Institutes of Health and additional federal partners under award numbers U01DA041022, U01DA041028, U01DA041048, U01DA041089, U01DA041106, U01DA041117, U01DA041120, U01DA041134, U01DA041148, U01DA041156, U01DA041174, U24DA041123, and U24DA041147. A full list of supporters is available at https://abcdstudy.org/nihcollaborators. A listing of participating sites and a complete listing of the study investigators can be found at https://abcdstudy.org/principal-investigators.html. ABCD consortium investigators designed and implemented the study and/or provided data but did not necessarily participate in analysis or writing of this report. This paper reflects the views of the authors and may not reflect the opinions or views of the NIH or ABCD consortium investigators.

The ABCD data repository grows and changes over time. The ABCD imaging data used in this report came from ABCD Collection 3165 (http://nda.nih.gov/edit_collection.html?id=3165). The SM data were obtained from the NDA ABCD RDS release (http://doi.org/10.15154/1504431).

Additonal data are provided in electronic supplementary material [47].

Authors' contributions. N.G.: conceptualization, data curation, formal analysis, software, visualization, writing—original draft, writing—review and editing; D.M.: conceptualization, data curation, formal analysis, methodology, resources, software, supervision, visualization, writing—original draft, writing—review and editing; P.A.B.: funding acquisition, writing—review and editing; E.S.F.: supervision, writing—original draft, writing—review and editing; A.G.T.: conceptualization, funding acquisition, methodology, project administration, supervision, writing—original draft, writing—review and editing. All authors gave final approval for publication and agreed to be held accountable for the work performed therein.

Competing interests. We declare we have no competing interests.

Funding. Authors N.G., D.M. and A.G.T. are supported by the NIMH Intramural Program ZICMH002960, P.A.B. is supported by the NIMH Intramural Program 1ZIAMH002783, and E.S.F. left the intramural program in 2020 to become an independent investigator at Dartmouth College with support from the NIMH Extramural Program R00MH120257.

Acknowledgements. The authors wish to acknowledge Eric Earl for assistance with figure creation, Regina Nuzzo for assistance with manuscript preparation, Stephen Smith for sharing the code from the original study and Anderson Winkler for assistance with permutation testing and CCA analysis. This work used the computational resources of the NIH HPC Biowulf cluster (http://hpc.nih.gov).

# Appendix A. Subject Measures

The final 74 subject measures used as input to the CCA were:

*nihtbx_picture_uncorrected, nihtbx_picture_agecorrected, nihtbx_cardsort_uncorrected, nihtbx_cardsort_ agecorrected, nihtbx_flanker_uncorrected, nihtbx_flanker_agecorrected, nihtbx_reading_uncorrected, nihtbx_reading_agecorrected, nihtbx_picvocab_uncorrected, nihtbx_picvocab_agecorrected, nihtbx_ pattern_uncorrected, nihtbx_pattern_agecorrected, nihtbx_list_uncorrected, nihtbx_list_agecorrected, fhx_ss_momdad_depression.bl, fhx_ss_parent_depression.bl, fhx_ss_momdad_drugs.bl, fhx_ss_parent_drugs.bl, fhx_ss_momdad_alc.bl, fhx_ss_parent_alc.bl, nihtbx_fluidcomp_fc, pea_wiscv_trs, pea_wiscv_tss, lmt_scr_num_correct, lmt_scr_rt_correct, lmt_scr_num_wrong, pea_ravlt_sd_listb_tc, resiliency_6a, resiliency_6b, fhx_ss_momdad_trouble.bl, fhx_ss_parent_trouble.bl, fhx_ss_momdad_nerves.bl, fhx_ss_ parent_nerves.bl, fhx_ss_momdad_suicide.bl, fhx_ss_parent_suicide.bl, cbcl_scr_syn_anxdep_r, cbcl_scr_ syn_anxdep_t, cbcl_scr_syn_withdep_r, cbcl_scr_syn_withdep_t, cbcl_scr_syn_somatic_r, cbcl_scr_syn_ somatic_t, cbcl_scr_syn_thought_r, cbcl_scr_syn_thought_t, cbcl_scr_syn_attention_r, cbcl_scr_syn_ attention_t, cbcl_scr_syn_aggressive_r, cbcl_scr_syn_aggressive_t, cbcl_scr_syn_rulebreak_r, cbcl_scr_ syn_rulebreak_t, cbcl_scr_syn_internal_r, cbcl_scr_syn_internal_t, cbcl_scr_syn_external_r, cbcl_scr_syn_ external_t, cbcl_scr_syn_totprob_r, cbcl_scr_syn_totprob_t, cbcl_scr_dsm5_depress_r, cbcl_scr_dsm5_ depress_t, cbcl_scr_dsm5_anxdisord_r, cbcl_scr_dsm5_anxdisord_t, cbcl_scr_dsm5_somaticpr_r, cbcl_scr_dsm5_somaticpr_t, cbcl_scr_dsm5_adhd_r, cbcl_scr_dsm5_adhd_t, cbcl_scr_dsm5_conduct_t, snellen_va, snellen_aid, asr_scr_anxdisord_r, fhx_ss_momdad_hospitalized.bl, fhx_ss_parent_hospitalized.bl, fhx_ss_momdad_professional.bl, fhx_ss_parent_professional.bl, neurocog_pc1.bl, neurocog_pc2.bl, neurocog_ pc3.bl.*

Descriptions of these 74 SMs are available in the 'Final 74 SMs' sheet of electronic supplementary material, file 2.

The 7 confound SMs which were regressed out were:

Scanner site (*abcd_site*), MRI scanner manufacturer (*mri_info_manufacturer*), mean frame displacement (*remaining_frame_mean_FD*), BMI (*anthro_bmi_calc*), Weight (*antho_weight_calc*), cube-root of total brain volume (*smri_vol_subcort.aseg_wholebrain*), cube-root of total intracranial volume (*smri_vol_subcort. aseg_intracranialcolume*).

The following 4 SMs were used in the positive–negative axis (in addition to the 74 SMs fed into the CCA); the SM variable name is given in parentheses:

BMI (*anthro_bmi_calc*), age (*age*), household income level (*household.income.bl*), parent education level (*high.educ*).

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
