## [Peer Review File · Royal Society Open Science]

Review History

RSOS-192086.R0 (Original submission)

Review form: Reviewer 1

Do you have any ethical concerns with this paper?

No

Have you any concerns about statistical analyses in this paper?

No

Recommendation?

Accept with minor revision

Comments to the Author(s)

Nikhil and colleagues propose a replication and extension of Smith *et al.*'s 2015 paper which found functional connectivity was related to a single axis of positive and negative phenotypes.

This is one of the most stark and notable neuroscience findings to have emerged in the past few years, and as noted by the authors no direct replication has been performed. The authors intention to replicate and extended this key study is both needed and likely to be of great interest. As noted by the authors, the code for reproducing Smith et al, is freely available so I am confident that the basic methodology and framework is the same. However, there are several areas I would appreciate seeing some greater detail and commentary on.

1. I was a little unclear as to if the 200-dimension group-ICA functional parcellation in the ABCD dataset is the exact same parcellation as used in the HCP, or if ICA had been rerun using the ABCD rs-fMRI data (I am assuming that a new ICA is being conducted). Could the authors please make this more salient in the text?

2. One page 5, line 36, the authors say their method is “similar” to what Smith et al used. I would like some clarification on what is meant by similar, here as similar does not mean exactly the same (if it is the exact same method but only with a different parcellation and dataset then say that, will help with clarity). On page 7, line 10 “similar” is also used, can this also be more exactly quantified please.

3. In page 5, line 27 the authors note a difference in the datasets are that the HCP used an ICA derived parcellation which the ABCD uses the Gordon. However a main point of the paper is that the Gordon parcellation is being applied to the HCP and an ICA (given a new one is actually being run, see comment 1) is being done on the ABCD. While it may be that these datasets intrinsically supply different parcellations, this is a moot point, because ultimately the same parcellation is being used in each dataset. I would revise this section to only mention differences that are not directly being tested/controlled for (and which could affect the results).

4. Could the authors please mention how the Gordon parcellation was (or how the parcellation intends to be) applied to the HCP-500 dataset? For instance was it warped from an MNI template into subject space, was it projected from one surface onto another etc?

5. In the main text itself, please indicate how the code and data for this study will be made available.

6. Could the exact number of subject measures be provided, instead of just the categories, please? Also, a breakdown of how many measures are included per category would be worthwhile.

7. Smith et al’s (2015) study did not perform a CCA on all connections and subject measures but instead used a principal component analysis to extract the first 100 components for both domains and used those instead. Could the authors please confirm they will be using this exact same approach? This point relates to my previous comment about needing to know many subject measures there are for the ABCD dataset, because the authors will not be able to perform exactly the same procedure if there are less than 100 measures. If indeed there are not 100 subject measures for the ABCD data, can the authors indicate how/if they intend to perform a PCA?

8. Related to comment 7 (and also following on from comment 2), I think the paper would benefit from having a section where they briefly describe the major steps of how exactly the CCA is performed (like how dimensionality reduction is being performed, confounds, how the permutations were calculated etc). This will help give readers confidence this replication has been done as close to Smith et al as possible. Also, could the authors please state the exact outcome measures they will be using (e.g. r value, extracting subject measures with the strongest associations, plotting connections with the strongest CCA weights etc)? The closest I can see is on page 5, line 36, where the authors state they will look for a “statistically significant CCA mode relative to a null distribution via permutation testing”, but I would like some more detail (if the

authors are only intending to use this single outcome to judge replication success, I would advise they also consider replicating other results Smith et al show).

9. Because the HCP data contains related individuals, their original permutation procedure kept this family structure intact. I would guess then this is not needed for the ABCD as most participants will be unrelated? Again, I think it is very beneficial for the paper if it can be shown clearly and exactly how the CCA approach here differed from Smith et al's.

10. The authors mention there are approximately 500 participants in the HCP-500 dataset and approximately 9,000 in the ABCD dataset (and there are around 1200 in the HCP-1200). However, as the authors note these datasets have been released so the exact number of participants should be known. Even if some participants need to be excluded due to outliers, missing data etc, could the number of potential participants please be provided?

11. Also on exclusion criteria, the authors mention missing data will be replaced with zeros but then in the next sentence note subjects with missing data were dropped (page 6, line 51). Can the authors make clear what/how much missing data for a participant would warrant replacement or exclusion? Additionally, I would like to see the exact exclusion criteria (e.g. how much missing data, what constitutes an outlier etc) mentioned in full in the paper, because these criteria are critical.

12. The authors note a train-test cross validation procedure will be used. Please state what proportion of participants will make up the training dataset and what proportion will make up the testing dataset?

13. I agree with the authors that using a developmental cohort is a strength as it allows assessment of whether Smith et al's original findings generalise to other groups. However, is it possible that because participants are likely developing at different rates, there is a heightened level of variation in the ABCD data and could this bias the results? Could this be examined in some way?

14. The final paragraph (page 7, line 7) can use some restructuring to make it clearer. First, break it into two paragraphs, one focusing on parcellations and the other based on how functional connectivity (FC) is calculated. Second, make it clear how the ICA in the ABCD data is being done (see comment 1). Finally, for the paragraph on FC calculation I was left a little confused as it made it sound like the main analyses (i.e. the computational and conceptual replication) are using different FC calculations, when they are in fact using the same one. Can this be rewritten so it is much clearer that this secondary analysis is being done using a simple Pearson correlation?

15. On page 3 line 22 and page 4 line 15, it says participants aged between 9-10 are used but on page 5, line 21 it says participants are aged between 9-11. Could this discrepancy be fixed please.

16. The abstract needs to be slightly reworked in my opinion, primarily to make it clear the focus is on networks as currently there is only one sentence which mentions networks. The term "functional brain activity" is used a lot, but to me this just sounds like a standard fMRI contrast. I would revise it to "functional brain connectivity", "functional connectivity", "functional brain network activity" or some variant thereof. In light of this I would change the order of the opening sentences of the abstract so it flows more logically. Something like "Understanding how brain functional activity is linked to human behaviour is a major goal of neuroscience. By modelling brain activity data as networks, known as functional connectivity, researchers can leverage the

mathematical tools of graph theory to probe these relationships. Numerous studies have investigated the relationship between functional connectivity and.....". Something like that.

17. The abstract should also mention the effect of parcellation is being examined as it is a big novelty of the study!

18. Following from this, in the introduction of the paper itself I would include a brief discussion of how a network is constructed, much like the abstract does, to just help orient the reader a bit better (especially because in the introduction the idea of networks is already bought up).

19. On page 3, line 55 the term "graph network" is used. These words are basically synonymous with each other, just use network.

20. In the third paragraph in the section on network neuroscience where parcellations are discussed for the first time (page 3), this opportunity should be used to discuss (even if just one sentence) that different parcellations can get different results (see Zalesky et al., 2010; Sala-Llonch et al., 2019).

21. Relating to my previous comment on page 4 line 47, it is mentioned in parenthesis that the type of parcellation can affect the result. This comes across as a tangential point when it really deserves more attention to help properly establish a rationale for why a different parcellation is being considered. Also in this part there is a hyperlink to Sala-Llonch et al. 2019 study. Please make sure to cite them properly, they did a lot of hard work and deserve a full citation!

The following comments are more suggestions and I am happy for the authors to state that addressing these is beyond the scope of the paper, but I would suggest the authors give it some consideration as I think it might help the paper be more impactful:

22. A novel aspect of this replication is the proposal to use an a priori parcellation (Gordon) over one derived from ICA. Given the effect of parcellation on network properties is well known, I would also suggest that the authors use this opportunity to examine the effects of other parcellations (for ease, this can just be restricted to the HCP data as I imagine there are subtleties in applying an adult parcellation of adolescent brains). For example the recently defined Schaefer parcellation (Schaefer et al., 2018) has received great interest as of late, and is also defined at a number of different nodal resolutions which would potentially allow for some extra exploration of how the number of nodes may affect the positive-negative axis (all the code/parcellations are freely available). However, at a minimum I strongly encourage the use of at least one other a priori parcellation as this would help make stronger statements about the effect of defining a parcellation a priori or a posteriori.

23. Additionally, to further drive home points about parcellation usage, including an anatomical parcellation (even if it is just the standard 34 cortical node one you get from Freesurfer) and seeing how that performs might be valuable. While this result will likely show weaker correlations than using a functional parcellation, many papers do analyse fMRI data using an anatomical parcellation so this might be a good opportunity to add more nails to that coffin.

24. As a final general comment, the authors may wish to read Wang et al., 2018 (available as a preprint currently), who has written an entire paper on the use of CCA in neuroimaging/biomedicine. It may offer some useful commentary on interpreting the CCA or dealing with this kind of data.

In summary the main aspect I wish to see addressed is a more detailed discussion of how the CCA itself is performed. However, I believe this and most of my other suggestions are minor and can be easily addressed.

Review references

Sala-Llonch, R., Smith, S. M., Woolrich, M., & Duff, E. P. (2019). Spatial parcellations, spectral filtering, and connectivity measures in fMRI: Optimizing for discrimination. *Human Brain Mapping*, 40(2), 407–419. <https://doi.org/10.1002/hbm.24381>

Schaefer, A., Kong, R., Gordon, E. M., Laumann, T. O., Zuo, X.-N., Holmes, A. J., ... Yeo, B. T. T. (2018). Local-Global Parcellation of the Human Cerebral Cortex from Intrinsic Functional Connectivity MRI. *Cerebral Cortex*, 28(9), 3095–3114. <https://doi.org/10.1093/cercor/bhx179>

Wang, H. T., Smallwood, J., Mourao-Miranda, J., Xia, C. H., Satterthwaite, T. D., Bassett, D. S., & Bzdok, D. (2018). Finding the needle in high-dimensional haystack: A tutorial on canonical correlation analysis. *arXiv preprint arXiv:1812.02598*.

Zalesky, A., Fornito, A., Harding, I. H., Cocchi, L., Yücel, M., Pantelis, C., & Bullmore, E. T. (2010). Whole-brain anatomical networks: Does the choice of nodes matter? *NeuroImage*, 50(3), 970–983. <https://doi.org/10.1016/j.neuroimage.2009.12.027>

Review form: Reviewer 2 (Angela R. Laird)

Do you have any ethical concerns with this paper?

No

Have you any concerns about statistical analyses in this paper?

No

Recommendation?

Major revision

Comments to the Author(s)

See attached file (Appendix A).

Review form: Reviewer 3

Do you have any ethical concerns with this paper?

No

Have you any concerns about statistical analyses in this paper?

No

Recommendation?

Accept with minor revision

Comments to the Author(s)

From section 4.3: I think it would be good to be slightly more precise about how the CCA output(s) will be compared against the original results - I would say it is not just important to show whether one or more CCA components are found, but how similar they are in terms of the functional networks and non-imaging-variables identified.

The paper is too vague about what data was used from ABCD and what preprocessing was applied. Were structured artefacts removed? Ideally to best match HCP preprocessing, ICA cleanup should be used, or an alternative that has been validated to be similarly effective. Was global signal regressed out? Hopefully not. Finally, the implication is that this was all surface/CIFTI-based analysis but this needs to be described more explicitly.

Decision letter (RSOS-192086.R0)

27-Jan-2020

Dear Mr Goyal,

The Editors assigned to your Stage 1 Replication submission ("The positive-negative mode link between brain connectivity, demographics and behavior: A pre-registered, replication of Smith et al.'s (2015)") have now received comments from reviewers. We would like you to revise your paper in accordance with the referee and editors suggestions which can be found below (not including confidential reports to the Editor). Please note this decision does not guarantee eventual acceptance.

Please submit a copy of your revised paper within three months, with a deadline of Monday 27 April 2020. If deemed necessary by the Editors, your manuscript will be sent back to one or more of the original reviewers for assessment. If the original reviewers are not available we may invite new reviewers.

When submitting your revised manuscript, you must respond to the comments made by the referees and upload a file "Response to Referees" in the "File Upload" step. Please use this to document how you have responded to the comments, and the adjustments you have made. In order to expedite the processing of the revised manuscript, please be as specific as possible in your response.

Once again, thank you for submitting your manuscript to Royal Society Open Science and we look forward to receiving your revision. If you have any questions at all, please do not hesitate to get in touch. Full author guidelines may be found at <https://royalsocietypublishing.org/rsos/replication-studies#AuthorsGuidance>.

Kind regards,
Lianne Parkhouse
Editorial Coordinator
Royal Society Open Science
openscience@royalsociety.org

on behalf of Professor Chris Chambers (Registered Reports Editor, Royal Society Open Science)
openscience@royalsociety.org

Editor Comments to Author (Professor Chris Chambers):

Three expert reviewers have now assessed the Stage 1 manuscript. All reviewers are positive in principle about the value of the replication attempt, but Reviewer 1 (and Reviewer 2 especially) are deeply critical of a wide range of aspects of the manuscript, including the degree of methodological detail, clarity of details that are provided, structure and coherence of the introduction, and overall validity of the study. For a regular manuscript, this combination of reviews would lead to outright rejection, but the advantage of the Replication format is that it provides the opportunity for reviewers to constructively guide authors toward addressing major (even severe) concerns, provided it is possible for the authors to do so.

On this basis, I would like to invite the authors to revise the manuscript but must add a caution that anything less than a comprehensive revision (and in places a significant rewrite) is likely to lead to rejection. Note that while Reviewer 3 judged both primary Stage 1 criteria to be met pending clarification of the points raised, Reviewer 1 and 2 judged at least one of the criteria to be unmet. Thus there is significant work to be done to bring the submission up to the point of achieving Stage 1 acceptance.

Reviewer Comments to Author:

Reviewer: 1

Comments to the Author(s)

Nikhil and colleagues propose a replication and extension of Smith et al's 2015 paper which found functional connectivity was related to a single axis of positive and negative phenotypes. This is one of the most stark and notable neuroscience findings to have emerged in the past few years, and as noted by the authors no direct replication has been performed. The authors intention to replicate and extended this key study is both needed and likely to be of great interest.

As noted by the authors, the code for reproducing Smith et al, is freely available so I am confident that the basic methodology and framework is the same. However, there are several areas I would appreciate seeing some greater detail and commentary on.

1. I was a little unclear as to if the 200-dimension group-ICA functional parcellation in the ABCD dataset is the exact same parcellation as used in the HCP, or if ICA had been rerun using the ABCD rs-fMRI data (I am assuming that a new ICA is being conducted). Could the authors please make this more salient in the text?
2. One page 5, line 36, the authors say their method is "similar" to what Smith et al used. I would like some clarification on what is meant by similar, here as similar does not mean exactly the same (if it is the exact same method but only with a different parcellation and dataset then say that, will help with clarity). On page 7, line 10 "similar" is also used, can this also be more exactly quantified please.
3. In page 5, line 27 the authors note a difference in the datasets are that the HCP used an ICA derived parcellation which the ABCD uses the Gordon. However a main point of the paper is that the Gordon parcellation is being applied to the HCP and an ICA (given a new one is actually being run, see comment 1) is being done on the ABCD. While it may be that these datasets intrinsically supply different parcellations, this is a moot point, because ultimately the same parcellation is being used in each dataset. I would revise this section to only mention differences that are not directly being tested/controlled for (and which could affect the results).
4. Could the authors please mention how the Gordon parcellation was (or how the parcellation intends to be) applied to the HCP-500 dataset? For instance was it warped from an MNI template into subject space, was it projected from one surface onto another etc?

5. In the main text itself, please indicate how the code and data for this study will be made available.
6. Could the exact number of subject measures be provided, instead of just the categories, please? Also, a breakdown of how many measures are included per category would be worthwhile.
7. Smith et al's (2015) study did not perform a CCA on all connections and subject measures but instead used a principal component analysis to extract the first 100 components for both domains and used those instead. Could the authors please confirm they will be using this exact same approach? This point relates to my previous comment about needing to know many subject measures there are for the ABCD dataset, because the authors will not be able to perform exactly the same procedure if there are less than 100 measures. If indeed there are not 100 subject measures for the ABCD data, can the authors indicate how/if they intend to perform a PCA?
8. Related to comment 7 (and also following on from comment 2), I think the paper would benefit from having a section where they briefly describe the major steps of how exactly the CCA is performed (like how dimensionality reduction is being performed, confounds, how the permutations were calculated etc). This will help give readers confidence this replication has been done as close to Smith et al as possible. Also, could the authors please state the exact outcome measures they will be using (e.g. r value, extracting subject measures with the strongest associations, plotting connections with the strongest CCA weights etc)? The closest I can see is on page 5, line 36, where the authors state they will look for a "statistically significant CCA mode relative to a null distribution via permutation testing", but I would like some more detail (if the authors are only intending to use this single outcome to judge replication success, I would advise they also consider replicating other results Smith et al show).
9. Because the HCP data contains related individuals, their original permutation procedure kept this family structure intact. I would guess then this is not needed for the ABCD as most participants will be unrelated? Again, I think it is very beneficial for the paper if it can be shown clearly and exactly how the CCA approach here differed from Smith et al's.
10. The authors mention there are approximately 500 participants in the HCP-500 dataset and approximately 9,000 in the ABCD dataset (and there are around 1200 in the HCP-1200). However, as the authors note these datasets have been released so the exact number of participants should be known. Even if some participants need to be excluded due to outliers, missing data etc, could the number of potential participants please be provided?
11. Also on exclusion criteria, the authors mention missing data will be replaced with zeros but then in the next sentence note subjects with missing data were dropped (page 6, line 51). Can the authors make clear what/how much missing data for a participant would warrant replacement or exclusion? Additionally, I would like to see the exact exclusion criteria (e.g. how much missing data, what constitutes an outlier etc) mentioned in full in the paper, because these criteria are critical.
12. The authors note a train-test cross validation procedure will be used. Please state what proportion of participants will make up the training dataset and what proportion will make up the testing dataset?
13. I agree with the authors that using a developmental cohort is a strength as it allows assessment of whether Smith et al's original findings generalise to other groups. However, is it possible that because participants are likely developing at different rates, there is a heightened

level of variation in the ABCD data and could this bias the results? Could this be examined in some way?

14. The final paragraph (page 7, line 7) can use some restructuring to make it clearer. First, break it into two paragraphs, one focusing on parcellations and the other based on how functional connectivity (FC) is calculated. Second, make it clear how the ICA in the ABCD data is being done (see comment 1). Finally, for the paragraph on FC calculation I was left a little confused as it made it sound like the main analyses (i.e. the computational and conceptual replication) are using different FC calculations, when they are in fact using the same one. Can this be rewritten so it is much clearer that this secondary analysis is being done using a simple Pearson correlation?

15. On page 3 line 22 and page 4 line 15, it says participants aged between 9-10 are used but on page 5, line 21 it says participants are aged between 9-11. Could this discrepancy be fixed please.

16. The abstract needs to be slightly reworked in my opinion, primarily to make it clear the focus is on networks as currently there is only one sentence which mentions networks. The term "functional brain activity" is used a lot, but to me this just sounds like a standard fMRI contrast. I would revise it to "functional brain connectivity", "functional connectivity", "functional brain network activity" or some variant thereof. In light of this I would change the order of the opening sentences of the abstract so it flows more logically. Something like "Understanding how brain functional activity is linked to human behaviour is a major goal of neuroscience. By modelling brain activity data as networks, known as functional connectivity, researchers can leverage the mathematical tools of graph theory to probe these relationships. Numerous studies have investigated the relationship between functional connectivity and.....". Something like that.

17. The abstract should also mention the effect of parcellation is being examined as it is a big novelty of the study!

18. Following from this, in the introduction of the paper itself I would include a brief discussion of how a network is constructed, much like the abstract does, to just help orient the reader a bit better (especially because in the introduction the idea of networks is already brought up).

19. On page 3, line 55 the term "graph network" is used. These words are basically synonymous with each other, just use network.

20. In the third paragraph in the section on network neuroscience where parcellations are discussed for the first time (page 3), this opportunity should be used to discuss (even if just one sentence) that different parcellations can get different results (see Zalesky et al., 2010; Sala-Llonch et al., 2019).

21. Relating to my previous comment on page 4 line 47, it is mentioned in parenthesis that the type of parcellation can affect the result. This comes across as a tangential point when it really deserves more attention to help properly establish a rationale for why a different parcellation is being considered. Also in this part there is a hyperlink to Sala-Llonch et al's 2019 study. Please make sure to cite them properly, they did a lot of hard work and deserve a full citation!

The following comments are more suggestions and I am happy for the authors to state that addressing these is beyond the scope of the paper, but I would suggest the authors give it some consideration as I think it might help the paper be more impactful:

22. A novel aspect of this replication is the proposal to use an a priori parcellation (Gordon) over one derived from ICA. Given the effect of parcellation on network properties is well known, I would also suggest that the authors use this opportunity examine the effects of other parcellations (for ease, this can just be restricted to the HCP data as I imagine there are subtleties in applying an adult parcellation of adolescent brains). For example the recently defined Schaefer parcellation (Schaefer et al., 2018) has received great interest as of late, and is also defined at a number of different nodal resolutions which would potentially allow for some extra exploration of how the number of nodes may affect the positive-negative axis (all the code/parcellations are freely available). However, at a minimum I strongly encourage the use of at least one other a priori parcellation as this would help make stronger statements about the effect of defining a parcellation a priori or a posteriori.

23. Additionally, to further drive home points about parcellation usage, including an anatomical parcellation (even if it is just the standard 34 cortical node one you get from Freesurfer) and seeing how that performs might be valuable. While this result will likely show weaker correlations than using a functional parcellation, many papers do analyse fMRI data using an anatomical parcellation so this might be a good opportunity to add more nails to that coffin.

24. As a final general comment, the authors may wish to read Wang et al., 2018 (available as a preprint currently), who has written an entire paper on the use of CCA in neuroimaging/biomedicine. It may offer some useful commentary on interpreting the CCA or dealing with this kind of data.

In summary the main aspect I wish to see addressed is a more detailed discussion of how the CCA itself is performed. However, I believe this and most of my other suggestions are minor and can be easily addressed.

Review references

Sala-Llonch, R., Smith, S. M., Woolrich, M., & Duff, E. P. (2019). Spatial parcellations, spectral filtering, and connectivity measures in fMRI: Optimizing for discrimination. *Human Brain Mapping*, 40(2), 407–419. <https://doi.org/10.1002/hbm.24381>

Schaefer, A., Kong, R., Gordon, E. M., Laumann, T. O., Zuo, X.-N., Holmes, A. J., ... Yeo, B. T. T. (2018). Local-Global Parcellation of the Human Cerebral Cortex from Intrinsic Functional Connectivity MRI. *Cerebral Cortex*, 28(9), 3095–3114. <https://doi.org/10.1093/cercor/bhx179>

Wang, H. T., Smallwood, J., Mourao-Miranda, J., Xia, C. H., Satterthwaite, T. D., Bassett, D. S., & Bzdok, D. (2018). Finding the needle in high-dimensional haystack: A tutorial on canonical correlation analysis. *arXiv preprint arXiv:1812.02598*.

Zalesky, A., Fornito, A., Harding, I. H., Cocchi, L., Yücel, M., Pantelis, C., & Bullmore, E. T. (2010). Whole-brain anatomical networks: Does the choice of nodes matter? *NeuroImage*, 50(3), 970–983. <https://doi.org/10.1016/j.neuroimage.2009.12.027>

Reviewer: 2

Comments to the Author(s)
see attached file

Reviewer: 3

Comments to the Author(s)

From section 4.3: I think it would be good to be slightly more precise about how the CCA output(s) will be compared against the original results - I would say it is not just important to show whether one or more CCA components are found, but how similar they are in terms of the functional networks and non-imaging-variables identified.

The paper is too vague about what data was used from ABCD and what preprocessing was applied. Were structured artefacts removed? Ideally to best match HCP preprocessing, ICA cleanup should be used, or an alternative that has been validated to be similarly effective. Was global signal regressed out? Hopefully not. Finally, the implication is that this was all surface/CIFTI-based analysis but this needs to be described more explicitly.

Author's Response to Decision Letter for (RSOS-192086.R0)

See Appendix B.

RSOS-192086.R1 (Revision)

Review form: Reviewer 1

Do you have any ethical concerns with this paper?

No

Have you any concerns about statistical analyses in this paper?

No

Recommendation?

Accept in principle

Comments to the Author(s)

The paper has been greatly improved upon since the initial revision. It is a lot more focused now and it is clearer what is being performed. In particular I think the rationale is a lot more compelling. All of my issues from the initial review have been more than adequately addressed!

Super minor thing: On page 4 of the manuscript, lines 43-49, these two sentences start the exact same way ("Prior to filtering"). In the spirit of good English please avoid this repetition.

I look forward to seeing the results! :)

Review form: Reviewer 2 (Angela R. Laird)

Do you have any ethical concerns with this paper?

No

Have you any concerns about statistical analyses in this paper?

No

Recommendation?

Accept in principle

Comments to the Author(s)

The authors have done a remarkable job in essentially rewriting the original version of this stage 1 registered replication and addressing prior reviewer concerns. The revised manuscript is well-designed, thoughtful, and provides very clear details regarding the proposed plan for conceptually replicating the results of Smith et al. (2015). The revision now includes a clearer rationale for the replication, as well as specific criteria for what will constitute a successful replication. The description of the subject filtering for image and behavioral data is excellent. Thanks very much for being so responsive to my earlier comments, it is much appreciated!

Very minor comments:

- manuscript page 3, line 17, I suggest specifying that "finding the positive-negative axis in the baseline ABCD dataset would enable..." (since you are pointing to follow-up work with other time points)
- Table 1, replace headers with "Sex" instead of "Gender" and "Race/Ethnicity" instead of "Race"

Review form: Reviewer 3

Do you have any ethical concerns with this paper?

No

Have you any concerns about statistical analyses in this paper?

No

Recommendation?

Accept in principle

Comments to the Author(s)

This is greatly improved.

One concern is whether ABCD is close enough (particularly wrt age) to HCP for this work to be considered a *replication* study. I'm on the fence on that. I think it just depends on your definition of "replication". Maybe this should be called a "generalisation" study?

My other concern is: if the idea is to show that one and only one significant CCA mode is found, there is the problem of increased statistical sensitivity by using much higher number of subjects in this study. Obviously finding more than one significant mode could be just a function of increased N, and so that would not be "failure". A more subtle issue is that in PCA and CCA there can be arbitrary rotation amongst components, so that if more than one mode were found, the original result might validly be mixed up across them.

Decision letter (RSOS-192086.R1)

Dear Mr Goyal

On behalf of the Editors, I am pleased to inform you that your Manuscript RSOS-192086.R1 entitled "The positive-negative mode link between brain connectivity, demographics, and behavior: A pre-registered, replication of Smith et al. (2015)" deemed suitable for in-principle acceptance in Royal Society Open Science subject to minor revision in accordance with the referee and editor suggestions. Please find their comments at the end of this email.

The reviewers and handling editors have recommended publication, but also suggest some minor revisions to your manuscript. Therefore, I invite you to respond to the comments and revise your manuscript.

Please you submit the revised version of your manuscript within 7 days (i.e. by the 20-Jun-2020). If you do not think you will be able to meet this date please let me know immediately.

Full author guidelines can be found here <https://royalsocietypublishing.org/rsos/replication-studies#AuthorsGuidance>

Kind regards
Andrew Dunn
Royal Society Open Science
openscience@royalsociety.org

on behalf of Professor Chris Chambers
(Subject Editor, Royal Society Open Science)
openscience@royalsociety.org

Associate Editor Comments to Author (Professor Chris Chambers):

Associate Editor: 1

Comments to the Author:

The manuscript was returned to the three expert reviewers who assessed the original Stage 1 submission. All are positive and recommend in-principle acceptance. I would like to authors pass through the manuscript and make a few final minor revisions as suggested in the reviews. I agree with Reviewer 3 that this submission sits close to the line of a constituting a "replication", but I am satisfied that it sufficiently meets the definition under this article type. Once the authors have

made these final revisions (or explained why they are not, as they may disagree with some suggestions), then in-principle acceptance will be forthcoming without requiring further in-depth Stage 1 review.

Reviewer comments to Author:

Reviewer: 1

Comments to the Author(s)

The paper has been greatly improved upon since the initial revision. It is a lot more focused now and it is clearer what is being performed. In particular I think the rationale is a lot more compelling. All of my issues from the initial review have been more than adequately addressed!

Super minor thing: On page 4 of the manuscript, lines 43-49, these two sentences start the exact same way ("Prior to filtering"). In the spirit of good English please avoid this repetition.

I look forward to seeing the results! :)

Reviewer: 2

Comments to the Author(s)

The authors have done a remarkable job in essentially rewriting the original version of this stage 1 registered replication and addressing prior reviewer concerns. The revised manuscript is well-designed, thoughtful, and provides very clear details regarding the proposed plan for conceptually replicating the results of Smith et al. (2015). The revision now includes a clearer rationale for the replication, as well as specific criteria for what will constitute a successful replication. The description of the subject filtering for image and behavioral data is excellent. Thanks very much for being so responsive to my earlier comments, it is much appreciated!

Very minor comments:

- manuscript page 3, line 17, I suggest specifying that "finding the positive-negative axis in the baseline ABCD dataset would enable..." (since you are pointing to follow-up work with other time points)

- Table 1, replace headers with "Sex" instead of "Gender" and "Race/Ethnicity" instead of "Race"

Reviewer: 3

Comments to the Author(s)

This is greatly improved.

One concern is whether ABCD is close enough (particularly wrt age) to HCP for this work to be considered a *replication* study. I'm on the fence on that. I think it just depends on your definition of "replication". Maybe this should be called a "generalisation" study?

My other concern is: if the idea is to show that one and only one significant CCA mode is found, there is the problem of increased statistical sensitivity by using much higher number of subjects in this study. Obviously finding more than one significant mode could be just a function of increased N, and so that would not be "failure". A more subtle issue is that in PCA and CCA there can be arbitrary rotation amongst components, so that if more than one mode were found, the original result might validly be mixed up across them.

Author's Response to Decision Letter for (RSOS-192086.R1)

See Appendix C.

Decision letter (RSOS-201090.R0)

Dear Mr Goyal,

On behalf of the Editor, I am pleased to inform you that your Manuscript RSOS-201090 entitled "The positive-negative mode link between brain connectivity, demographics, and behavior: A pre-registered, replication of Smith et al. (2015)" has been accepted in principle for publication in Royal Society Open Science.

Please note that you must now register your approved protocol on the Open Science Framework (<https://osf.io/rr>), using the 'Submit your approved Registered Report' option and then the 'Registered Report Protocol Preregistration' option. Please use the Registered Report option even though your article is being accepted as a Stage 1 Replication. Further into the registration process, in the Journal Title field enter 'Royal Society Open Science (Replication article type, Preregistered track)'. Please note that a time-stamped, independent registration of the protocol is mandatory under journal policy, and manuscripts that do not conform to this requirement cannot be considered at Stage 2. The protocol should be registered unchanged from its current approved state. Please include a URL to the protocol in your Stage 2 manuscript (e.g. 'This article received in-principle acceptance (IPA) at Royal Society Open Science. Following IPA, the accepted Stage 1 version of the manuscript, not including results and discussion, was preregistered on the OSF (URL).')

Following completion of your study, we invite you to resubmit your paper for peer review as a Stage 2 Replication. Please note that your manuscript can still be rejected for publication at Stage 2 if the Editors consider any of the following conditions to be met:

- The Introduction and methods deviated from the approved Stage 1 submission (required).
- The authors' conclusions were not considered justified given the data.

We encourage you to read the complete guidelines for authors concerning Stage 2 submissions at : <https://royalsocietypublishing.org/rsos/replication-studies#AuthorsGuidance>. Please especially note the requirements for data sharing and that withdrawing your manuscript will result in publication of a Withdrawn Registration.

We encourage you to read the complete guidelines for authors concerning Stage 2 submissions at <https://royalsocietypublishing.org/rsos/registered-reports#ReviewerGuideRegRep>. Please especially note the requirements for data sharing and that withdrawing your manuscript will result in publication of a Withdrawn Registration.

Once again, thank you for submitting your manuscript to Royal Society Open Science and I look forward to receiving your Stage 2 submission. If you have any questions at all, please do not hesitate to get in touch. We look forward to hearing from you shortly with the anticipated submission date for your stage two manuscript.

on behalf of Professor Chris Chambers (Registered Reports Editor, Royal Society Open Science)
 openscience@royalsociety.org

Author's Response to Decision Letter for (RSOS-201090.R0)

See Appendix D.

RSOS-201090.R1 (Revision)

Review form: Reviewer 1

Is the manuscript scientifically sound in its present form?

Yes

Are the interpretations and conclusions justified by the results?

Yes

Is the language acceptable?

Yes

Do you have any ethical concerns with this paper?

No

Have you any concerns about statistical analyses in this paper?

No

Recommendation?

Accept with minor revision

Comments to the Author(s)

I had previously reviewed Goyal et al's preregistration and found that to be of very high quality after some gentle prodding by reviewers. In light of this, I will primarily focus on the results and discussion as we have been there and done that with the introduction and method. Pleasingly, these sections are already of a high standard, and I only have a few minor points.

In the abstract, could the authors list the three pre-registered criteria and indicate the ones which were supported? The third-last and second-last sentences could be cut down in order to make room for this additional information.

The criteria to gauge a resting-state scan as "good", was that derived from anywhere?

One thing that immediately jumps out to me from this result is the possibility that the positive-negative axis strengths across neurodevelopment, however I feel the authors do not confidently propose such a hypothesis. It is briefly mentioned on page 26, lines 43-45, but I think it would be

worth more clearly indicating why the results of this study support such a hypothesis/proposal for future work.

On page 28 lines 6-9 the following is stated: "This is consistent with the findings on the development of the default mode network and higher-order association cortices in general, such that these regions undergo a protracted development into early adulthood". I would expand on this a little and explicitly mention this is because default-mode connectivity is more fragmented/weakened in youth.

Review form: Reviewer 2 (Angela R. Laird)

Is the manuscript scientifically sound in its present form?

Yes

Are the interpretations and conclusions justified by the results?

Yes

Is the language acceptable?

Yes

Do you have any ethical concerns with this paper?

No

Have you any concerns about statistical analyses in this paper?

No

Recommendation?

Accept with minor revision

Comments to the Author(s)

This is an excellent Stage 2 manuscript that reports a conceptual replication of Smith et al.'s positive-negative axis and primary CCA mode findings in the ABCD dataset. The manuscript includes introduction and methods that are the same as what was approved at Stage 1, with some minor wording changes that only serve to provide additional clarification and improve phrasing. Given a complex series of analyses and a challenging dataset, I am especially appreciative of the clarity and presentation of the results. The protocol deviations are clearly detailed and well justified. I was happy to see the discussion point regarding potential developmental effects associated with DMN-related differences in Smith et al.'s results and those shown here, as I think that this is a critical aspect when considering any replication differences. In addition to the Sydnor et al. study, it would be helpful to also cite a recent study by Fan et al. that supports the idea that the observed clustering differences (i.e., "nodes spread out among different clusters") may be linked to reduced DMN connectivity in middle childhood (<https://pubmed.ncbi.nlm.nih.gov/33221440/>).

Also, I appreciate the caution in the discussion regarding over-interpreting the proxy measures for SES (e.g., education and income), but I also think it's worth an explicit mention that these are key factors of the social determinants of health (e.g., these are #1 and #2 on the WHO's list: https://www.who.int/health-topics/social-determinants-of-health#tab=tab_1). So when I look at Figure 4 and see the two gray variables, in my mind they are not outliers, but very well aligned with your conclusions about the positive-negative axis from a health-based perspective. An additional sentence in this paragraph on page 26 that points to the need for expanding

contextualizing variables related to the social determinants of health in future studies, to complement the fairly standard neurocognitive and clinical variables, would be a nice touch.

Minor points:

Abstract, line 32: access – > axis

Table 1: Race – > Race/Ethnicity

Review form: Reviewer 3

Is the manuscript scientifically sound in its present form?

Yes

Are the interpretations and conclusions justified by the results?

Yes

Is the language acceptable?

Yes

Do you have any ethical concerns with this paper?

No

Have you any concerns about statistical analyses in this paper?

No

Recommendation?

Accept as is

Comments to the Author(s)

I reviewed the original stages of this paper, and am very impressed with the work as a whole. This was clearly a large amount of work and the authors are to be congratulated. I have just one minor query: The authors rightly state that "... in PCA/CCA there can be an arbitrary rotation amongst components that can spread the original result validly across the multiple significant modes". Given that the authors found more than one significant CCA mode, I was surprised that they did not act on this concern, and apply ICA, to see how this changed the results - e.g. how this changed the similarity of components to the original Smith results? This might be added.

Decision letter (RSOS-201090.R1)

Dear Mr Goyal

On behalf of the Editor, I am pleased to inform you that your Stage 2 Replication submission RSOS-201090.R1 entitled "The positive-negative mode link between brain connectivity, demographics, and behavior: A pre-registered, replication of Smith et al. (2015)" has been

accepted for publication in Royal Society Open Science subject to minor revision in accordance with the referee suggestions. Please find the referees' comments at the end of this email.

The reviewers and Subject Editor have recommended publication, but also suggest some minor revisions to your manuscript. We invite you to respond to the comments and revise your manuscript. Below the referees' and Editors' comments (where applicable) we provide additional requirements. Final acceptance of your manuscript is dependent on these requirements being met. We provide guidance below to help you prepare your revision.

Please submit your revised manuscript and required files (see below) no later than 7 days from today's (ie 06-Dec-2021) date. Note: the ScholarOne system will 'lock' if submission of the revision is attempted 7 or more days after the deadline. If you do not think you will be able to meet this deadline please contact the editorial office immediately.

Kind regards,
Professor Chris Chambers
Royal Society Open Science
openscience@royalsociety.org

on behalf of Professor Chris Chambers (Registered Reports Editor, Royal Society Open Science)
openscience@royalsociety.org

Associate Editor Comments to Author (Professor Chris Chambers):

Associate Editor: 1

Comments to the Author:

The three reviewers from Stage 1 kindly returned to evaluate the Stage 2 manuscript. As you will see, their assessments are positive, noting only minor issues and some points of clarification to address in order to achieve full acceptance. Reviewer 3 suggests an additional analysis which could be useful. Note that authors of replications at RSOS are not required to include additional analyses that were not preregistered at Stage 1. You are, however, free to conduct and report this analysis if you wish. Provided you are able to respond thoroughly to the points raised in a revision then full acceptance should be forthcoming without requiring further in-depth review.

Reviewers' comments to Author:

Reviewer: 1

Comments to the Author(s)

I had previously reviewed Goyal et al's preregistration and found that to be of very high quality after some gentle prodding by reviewers. In light of this, I will primarily focus on the results and discussion as we have been there and done that with the introduction and method. Pleasingly, these sections are already of a high standard, and I only have a few minor points.

In the abstract, could the authors list the three pre-registered criteria and indicate the ones which were supported? The third-last and second-last sentences could be cut down in order to make room for this additional information.

The criteria to gauge a resting-state scan as “good”, was that derived from anywhere?

One thing that immediately jumps out to me from this result is the possibility that the positive-negative axis strengths across neurodevelopment, however I feel the authors do not confidently propose such a hypothesis. It is briefly mentioned on page 26, lines 43-45, but I think it would be worth more clearly indicating why the results of this study support such a hypothesis/proposal for future work.

On page 28 lines 6-9 the following is stated: “This is consistent with the findings on the development of the default mode network and higher-order association cortices in general, such that these regions undergo a protracted development into early adulthood”. I would expand on this a little and explicitly mention this is because default-mode connectivity is more fragmented/weakened in youth.

Reviewer: 2

Comments to the Author(s)

This is an excellent Stage 2 manuscript that reports a conceptual replication of Smith et al.’s positive-negative axis and primary CCA mode findings in the ABCD dataset. The manuscript includes introduction and methods that are the same as what was approved at Stage 1, with some minor wording changes that only serve to provide additional clarification and improve phrasing. Given a complex series of analyses and a challenging dataset, I am especially appreciative of the clarity and presentation of the results. The protocol deviations are clearly detailed and well justified. I was happy to see the discussion point regarding potential developmental effects associated with DMN-related differences in Smith et al.’s results and those shown here, as I think that this is a critical aspect when considering any replication differences. In addition to the Sydnor et al. study, it would be helpful to also cite a recent study by Fan et al. that supports the idea that the observed clustering differences (i.e., “nodes spread out among different clusters”) may be linked to reduced DMN connectivity in middle childhood (<https://pubmed.ncbi.nlm.nih.gov/33221440/>).

Also, I appreciate the caution in the discussion regarding over-interpreting the proxy measures for SES (e.g., education and income), but I also think it’s worth an explicit mention that these are key factors of the social determinants of health (e.g., these are #1 and #2 on the WHO’s list: https://www.who.int/health-topics/social-determinants-of-health#tab=tab_1). So when I look at Figure 4 and see the two gray variables, in my mind they are not outliers, but very well aligned with your conclusions about the positive-negative axis from a health-based perspective. An additional sentence in this paragraph on page 26 that points to the need for expanding contextualizing variables related to the social determinants of health in future studies, to complement the fairly standard neurocognitive and clinical variables, would be a nice touch.

Minor points:

Abstract, line 32: access → axis

Table 1: Race → Race/Ethnicity

Reviewer: 3

Comments to the Author(s)

I reviewed the original stages of this paper, and am very impressed with the work as a whole. This was clearly a large amount of work and the authors are to be congratulated. I have just one

minor query: The authors rightly state that "... in PCA/CCA there can be an arbitrary rotation amongst components that can spread the original result validly across the multiple significant modes". Given that the authors found more than one significant CCA mode, I was surprised that they did not act on this concern, and apply ICA, to see how this changed the results - e.g. how this changed the similarity of components to the original Smith results? This might be added.

===PREPARING YOUR MANUSCRIPT===

one version should clearly identify all the changes that have been made (for instance, in coloured highlight, in bold text, or tracked changes);

===PREPARING YOUR REVISION IN SCHOLARONE===

Please ensure that you include a summary of your paper at the 'Type, Title, & Abstract' step. This should be no more than 100 words to explain to a non-scientific audience the key findings of your research. This will be included in a weekly highlights email circulated by the Royal Society press

office to national UK, international, and scientific news outlets to promote your work. An effective summary can substantially increase the readership of your paper.

-- If you are requesting an article processing charge waiver, you must select the relevant waiver option (if requesting a discretionary waiver, the form should have been uploaded, see 'File upload' above).

-- If you have uploaded any electronic supplementary (ESM) files, please ensure you follow the guidance at <https://royalsociety.org/journals/authors/author-guidelines/#supplementary-material> to include a suitable title and informative caption. An example of appropriate titling and captioning may be found at https://figshare.com/articles/Table_S2_from_Is_there_a_trade-off_between_peak_performance_and_performance_breadth_across_temperatures_for_aerobic_scope_in_teleost_fishes_/3843624.

Author's Response to Decision Letter for (RSOS-201090.R1)

See Appendix E.

Decision letter (RSOS-201090.R2)

Dear Mr Goyal:

It is a pleasure to accept your Stage 2 Replication entitled "The positive-negative mode link between brain connectivity, demographics, and behavior: A pre-registered replication of Smith et al. (2015)" in its current form for publication in Royal Society Open Science. Congratulations on an excellent piece of work.

on behalf of Professor Chris Chambers (Subject Editor)
openscience@royalsociety.org

Appendix A

Stage 1 Primary Criterion #1: Have the authors provided a sufficiently clear and detailed description of the methods for another researcher to closely replicate the proposed experimental procedures and analysis pipeline, and to prevent undisclosed flexibility in the experimental procedures or analysis pipeline?

No. The Methods section is insufficient and does not allow for a full evaluation of the plans for this registered replication. The methods should include a clear description of all data and analyses to be performed; ascertaining many of the analytic details requires reading the original study, which shouldn't be necessary. Some of these details are found in other sections of the Introduction, which makes for a difficult reading experience. In terms of the datasets, it's great that the authors have provided the specific NDA collection numbers, as this provides a link to the exact datasets analyzed. However, these cannot be accessed without an executed NDA data use agreement, so most readers won't be able to simply look up the details of these collections. The manuscript did not include any other description of the data, not even sample size. The original Smith et al. (2015) study included 461 participants from the HCP 500 subject release. In the proposed replication, no description is provided as to what inclusion/exclusion criteria will be applied to the HCP 500, HCP 1200, and ABCD release 2.0 datasets, particularly those related to image quality metrics or data completeness. No description is provided of the imaging parameters, no description is provided of the behavioral measures. An Appendix A was referenced, but was not included in the manuscript submission. These details are critically important to evaluating the replication. In terms of the analyses, the parcellations and CCA analyses are loosely described elsewhere, so getting a sense of what the authors will actually do requires pulling bits of text from different places in the Introduction. A clear description of the overall workflow, presented solely in the Methods and not the Introduction, would make the

reader's experience much more enjoyable, and are needed for a complete evaluation of this work. Removing these analytic details from the Introduction will allow for a clearer presentation of the motivation and rationale (see below).

Stage 1 Primary Criterion #2: Does the manuscript describe a sufficiently valid (i.e. close) and robust (e.g. statistically powerful) replication of the original study methods and rationale to provide an indication of replicability?

No. The authors seek to replicate the results of Smith et al. (2015), both in the same dataset and an independent dataset, while testing for the effect of parcellation approach. While it's commendable that they're investigating the effects of parcellation and dataset on the analysis, the two datasets are so different from one another that generalization will be difficult to interpret. Three major factors that are not investigated in this design are participant age (see below), study site (esp. scanner), and behavioral metrics (insufficient details were provided on these). (As an aside, WUSTL is the HCP site, but is also an ABCD site, so there is a missed opportunity to analyze both HCP and ABCD data acquired at that site.) Other factors worth looking into include preprocessing pipeline, acquisition parameters, and sample demographics (beyond age), none of which were mentioned. Finally, it's unclear what would constitute a successful/failed replication in any of the analysis, which is a critical aspect of the plan that was not addressed. Sample size and power were not addressed.

Stage 1 Secondary Criterion #1: Are the proposed hypotheses logical and supported by a clear and persuasive scientific rationale?

No. I'll start with my concerns regarding the organization of the manuscript, as these are coupled with my concerns regarding

presentation of the scientific rationale for the proposed replication. Overall, although it was brief, I found the manuscript fairly challenging to read, and revising the organization and presentation of various pieces are likely to significantly improve the manuscript. Much of the Introduction is a broad and diffuse summary of connectomics, which could be removed in favor of a well-organized description of the rationale and motivation for the proposed replication. The initial citation of 7 studies relating brain and behavior seems to be hand-picked and does not have a clear purpose. The next paragraph is a summary of the study to be replicated (Smith et al., 2015), but the motivation for the replication presented here is simply to assess the robustness of the original finding. This is not a compelling argument, particularly in light of the age differences between the HCP and ABCD samples. From here, the section on “Network neuroscience” backtracks to a broad review of fMRI, with an abrupt transition to parcellations and graph theory. This text isn’t needed, and it detracts from what should be the purpose of the introduction: to set up the purpose of the replication. The final paragraph of section 2 is briefly describes the HCP and ABCD projects, but this seems out of place and insufficient. HCP inclusion of “sibships” is mentioned (without a definition), without similarly noting the inclusion of twin data in the ABCD sample.

From there, the introduction swings back to the original Smith et al. (2015) study and loosely describes the original methodology. It would be helpful if this section also included a summary of the significance of this finding. When the original Smith et al. (2015) study is introduced, identifying the significance of this finding will help in articulating the significance of the replication. I’m not someone who thinks that manuscripts should be rejected for “not being significant enough”, but I do think that authors should strive to translate data-driven findings into a statement that summarizes their significance or implications. This doesn’t appear to be addressed, but to be fair, I’m not certain that it was addressed in

the original publication (likely a function of the Brief Communication format). It's remarkable that "a single highly significant CCA mode that relates functional connectomes to subject measures" was observed, and that "nearly all the positively correlated SMs are commonly considered as positive personal qualities or indicators... and all negatively correlated SMs relate to negative traits". But beyond this finding itself, it's not clear how the result informs our understanding of human functional brain organization.

The manuscript then states that the motivation of the replication is to address the generalizability of the results with respect to the brain parcellation and the study sample (see above for comments on the challenges of addressing generalizability with this design). It is stated that the HCP sample is representative of the US population, but this term should be avoided, both with respect to HCP and ABCD samples (please see Compton, Dowling, and Garavan, *JAMA Pediatr*, DOI: 10.1001/jamapediatrics.2019.2081). Parcellation choice is an important aspect for this replication of Smith et al. (2015), and I agree with the authors that multiple parcellation approaches should be tested to address generalizability of the findings. However, a replication and extension that leverages both HCP and ABCD samples should include clear age-related questions, as this is a critical difference between these two study samples. Importantly, the ABCD release 2.0 baseline data includes youth aged 9-10 years old (the manuscript should be revised throughout to reflect that these are pre-adolescent, not adolescent, participants). Extending this replication of Smith et al. (2015) to ABCD data introduces a lifespan question that asks whether the observed feature of brain organization is present prior to the neurodevelopmental changes that occur during adolescence and early adulthood. In this respect, the manuscript does not at all address the age-related relevance of extending the analysis from HCP to ABCD samples. The authors "hypothesized that the single

CCA mode and positive-negative axis found in the original study would successfully replicate in the ABCD dataset, regardless of the parcellation and subject measures used”, but what is the justification for this hypothesis? What evidence suggests that this single CCA mode and positive-negative axis is present prior to adolescence?

Related to motivation and rationale, I found the presentation of the analytic strategy (page 5) to be less than compelling. It comes across as if the original choice of parcellation is based on convenience: the authors will employ the same parcellation as Smith et al. (2015) for the HCP data, and then use the Gordon parcellation for the ABCD data. As a reader, convenience is never a sufficient justification, so this raises concerns. But then the manuscript points to a figure (although not spelled out in the text) that indicates that multiple analyses will be performed such that both parcellations will be employed on both samples. This is an awkward presentation of a critical aspect of the replication. Moreover, the figure includes the HCP 500 and ABCD 2.0 analyses, but not the HCP 1200 analysis.

I am highly enthusiastic about registered replications, but I found this Stage 1 manuscript to be quite difficult to read and review. Replication of the Smith et al. (2015) finding is sound idea, and I applaud the first author for taking on such an ambitious project and making use of both HCP and ABCD datasets. However, the execution of this idea requires significant additional justification and will require additional consideration of age-related differences that are likely to impact generalizability. Overall, there are a number of details that need to be clarified and the manuscript requires extensive rewriting and additional oversight.

Stage 1 Secondary Criterion #2: Are the proposed procedures and analysis pipeline methodologically sound?

Insufficient details provided to fully evaluate.

Stage 1 Secondary Criterion #3: Have the authors considered sufficient outcome-neutral conditions (e.g. absence of floor or ceiling effects; positive controls; other quality checks) for ensuring that the results obtained are able to test the stated hypotheses?

No. It's unclear what would constitute a successful/failed replication in any of the analysis. Moreover, while parcellation and dataset will be tested, other important factors will not (e.g., age, study site, behavioral metrics, imaging parameters, preprocessing, etc.), which will likely contribute to ambiguity in results.

Do you have any concerns about the proposed statistical analyses in this study? If so, please specify them explicitly in your report.

None.

It is a condition of publication that authors make their supporting data, code and materials available. Is it clear from reading the paper how the authors will make these available?

The link to the GitHub repository should be included.

Comment on the Title

Please remove the second comma from the title (and add an Oxford comma after “demographics”). Also, the citation of the original study should not be possessive (i.e., it should be Smith et al. (2015) instead of Smith et al.’s).

Appendix B

We thank the reviewers for taking time to provide us detailed feedback on our initial manuscript submission. We have entirely rewritten the manuscript to clarify previously vague aspects of our study and address the concerns raised by the reviewers.

Major changes include:

1. This study no longer attempts to replicate the original study in both the HCP and ABCD datasets. Instead, it focuses entirely on conceptually replicating Smith et al. in the ABCD dataset. The computational replications in the HCP dataset used to validate our analysis pipeline have been documented in a preprint available on the bioRxiv (<https://www.biorxiv.org/content/10.1101/2020.04.23.058313v1>).
2. Since our initial submission, a second comprehensive survey of different parcellations schemes has been published by Pervaiz et al. (<https://doi.org/10.1016/j.neuroimage.2020.116604>) adding to the excellent work of Sala-Llonch et al. (2019) noted by reviewer 1. These papers demonstrate the importance of parcellation type and illustrate how involved a proper exploration of parcellation type needs to be. We have therefore chosen to remove the exploration of different parcellations of the HCP/ABCD data from our manuscript. We instead use only a 200-dimension group-ICA parcellation on the ABCD dataset. This is in line with both the original study and shown by both Sala-Llonch et al. and Pervaiz et al. to have strong predictive performance. We leave the exploration of parcellation on this particular result as a topic for future work.
3. A significant amount of preliminary analysis has been added to identify:
 - a. The number of subjects and Subject Measures (SMs) in the ABCD data release that are valid for our analysis after quantitative and qualitative exclusions
 - b. Demographics of the sample before and after filtering
 - c. A one-to-one matching of SMs
 - d. The exact steps needed to generate the group-ICA of ABCD resting-state data for valid subjects

Again, we thank the reviewers for the extremely helpful feedback provided and apologize for the shortcomings of our first submission. We hope that this comprehensive revision and our responses to the reviewers address all concerns and make our goals clear for this study. Point-by-point responses to the reviewers' concerns follows below.

Reviewer 1

- 1. I was a little unclear as to if the 200-dimension group-ICA functional parcellation in the ABCD dataset is the exact same parcellation as used in the HCP, or if ICA had been rerun using the ABCD rs-fMRI data (I am assuming that a new ICA is being conducted). Could the authors please make this more salient in the text?**

The reviewer is correct in that we are performing a group-ICA on the ABCD rs-fMRI data. This has been clarified in the text, in the introduction (1.3. *Conceptual replication*) and in the methods (2.3.3. *Group-ICA of ABCD resting-state fMRI data*).

- 2. One page 5, line 36, the authors say their method is “similar” to what Smith et al used. I would like some clarification on what is meant by similar, here as similar does not mean exactly the same (if it is the exact same method but only with a different parcellation and dataset then say that, will help with clarity). On page 7, line 10 “similar” is also used, can this also be more exactly quantified please.**

(Page 5, line 36): We have quantified the definition of a successful replication (see section 1.3.1. *Replication overview*) and clarified the null distribution statistical testing (2.4.2. *Canonical Correlation Analysis*).

(Page 7, line 10): The process of calculating the 200-dimension group ICA has been made explicit in the methods (2.3.3. *Group-ICA of ABCD resting-state fMRI data*)

- 3. In page 5, line 27 the authors note a difference in the datasets are that the HCP used an ICA derived parcellation which the ABCD uses the Gordon. However a main point of the paper is that the Gordon parcellation is being applied to the HCP and an ICA (given a new one is actually being run, see comment 1) is being done on the ABCD. While it may be that these datasets intrinsically supply different parcellations, this is a moot point, because ultimately the same parcellation is being used in each dataset. I would revise this section to only mention differences that are not directly being tested/controlled for (and which could affect the results).**

We have revised the manuscript to reflect that we are only calculating a 200-dimension group-ICA functional parcellation on the ABCD data. See our summary comments at the top of this document for details and rationale

Differences between the ABCD and HCP datasets (especially subject demographics, age, behavioral metrics, acquisition protocols, and preprocessing pipelines) which could affect results are discussed in 1.3. *Conceptual replication*. Major factors such as age, scanner type, scan site, weight, and BMI are regressed out of our analysis.

- 4. Could the authors please mention how the Gordon parcellation was (or how the parcellation intends to be) applied to the HCP-500 dataset? For instance was it warped from an MNI template into subject space, was it projected from one surface onto another etc?**

This aspect of our study has been removed. See our summary comments at the top of this document for details and rationale.

5. *In the main text itself, please indicate how the code and data for this study will be made available.*

A statement with a link to the project's *Open Science Foundation* page has been added to section 2.5. *Code Availability*. All code will be made available in this repository. The 2.0.1 Release of the ABCD dataset is available for download from the NIMH Data Archive. Two links are provided in the methods section:

https://nda.nih.gov/edit_collection.html?id=3165
<http://doi.org/10.15154/1504431>

6. *Could the exact number of subject measures be provided, instead of just the categories, please? Also, a breakdown of how many measures are included per category would be worthwhile.*

A one-to-one matching has been performed identifying 143 ABCD subject measures that either exactly or approximately match HCP subject measures. Of these, 89 passed quantitative filtering, and 74 are ultimately used in our analysis. A full listing of each subject measure used is now included as a supplementary file. This is discussed in detail in section 2.3. *Data Preprocessing*.

7. *Smith et al's (2015) study did not perform a CCA on all connections and subject measures but instead used a principal component analysis to extract the first 100 components for both domains and used those instead. Could the authors please confirm they will be using this exact same approach? This point relates to my previous comment about needing to know many subject measures there are for the ABCD dataset, because the authors will not be able to perform exactly the same procedure if there are less than 100 measures. If indeed there are not 100 subject measures for the ABCD data, can the authors indicate how/if they intend to perform a PCA?*

This aspect of our study has been clarified, with all data preprocessing, data preparation, and data analysis steps laid out in sections 2.2. *Data Acquisition*, 2.3. *Data Preprocessing*, and 2.4. *Data Analysis*.

Because there are only 74 SMs suitable for analysis, we perform a 70-dimension PCA for each domain (SM and connectivity). Although this is smaller than the 100-dimension PCA in the original study, Smith et al. noted that there were no statistical differences in the final CCA model when the pre-CCA reduction of SMs and netmats was run with a much smaller number of PCA components (specifically, 30 instead of 100).

8. *Related to comment 7 (and also following on from comment 2), I think the paper would benefit from having a section where they briefly describe the major steps of how exactly the CCA is performed (like how dimensionality reduction is being performed, confounds, how the permutations were calculated etc). This will help give readers confidence this replication has been done as close to Smith et al as possible.*

We apologize for confusion on this aspect of our study - the exact steps have been included in sections 2.3. *Data preprocessing* and 2.4. *Data Analysis*, as well as a figure (Figure 1) that depicts the processing pipeline leading up to the CCA.

Also, could the authors please state the exact outcome measures they will be using (e.g. r value, extracting subject measures with the strongest associations, plotting connections with the strongest CCA weights etc)? The closest I can see is on page 5, line 36, where the authors state they will look for a “statistically significant CCA mode relative to a null distribution via permutation testing”, but I would like some more detail (if the authors are only intending to use this single outcome to judge replication success, I would advise they also consider replicating other results Smith et al show).

We have clarified this aspect of the study in section 1.3.1. *Replication overview*. Exact criteria for successful replication of the primary CCA mode are given; in addition we have listed the post-hoc analyses we will perform: a hierarchical analysis of the connectomes, and which functional connections correlate the most with the primary CCA mode; a clustering analysis to identify the major functional networks of the brain in which the 200 nodes fall into; and an train-test split to test the predictive performance of the CCA model on unseen data. However, we do not define criteria that constitute a successful replication of the post-hoc analyses since they are exploratory.

9. *Because the HCP data contains related individuals, their original permutation procedure kept this family structure intact. I would guess then this is not needed for the ABCD as most participants will be unrelated? Again, I think it is very beneficial for the paper if it can be shown clearly and exactly how the CCA approach here differed from Smith et al’s.*

Clarification about the number of sibships (and a breakdown of the types) in our study (before and after subject exclusion) has been added to section 2.1. *Participants*. Permutation calculation is discussed in section 2.4.2. *Canonical Correlation Analysis*. In our final sample there are 1,339 subjects categorized as MZ/DZ twins, and 1,235 subjects classified as non-twin siblings. We use the same permutation tool (*hcp2blocks*) as in the original study and compute permutations in the exact same way.

10. *The authors mention there are approximately 500 participants in the HCP-500 dataset and approximately 9,000 in the ABCD dataset (and there are around 1200 in the HCP-1200). However, as the authors note these datasets have been released so the exact number of participants should be known. Even if some participants need to be excluded due to outliers, missing data etc, could the number of potential participants please be provided?*

An exact number of subjects (n=11,875 before exclusion, n=7,810 and after subject exclusion) has been added to 2.1. *Participants*. The process for filtering out subjects is explained in sections 2.3.1. and 2.3.2.

- 11. Also on exclusion criteria, the authors mention missing data will be replaced with zeros but then in the next sentence note subjects with missing data were dropped (page 6, line 51). Can the authors make clear what/how much missing data for a participant would warrant replacement or exclusion? Additionally, I would like to see the exact exclusion criteria (e.g. how much missing data, what constitutes an outlier etc) mentioned in full in the paper, because these criteria are critical.**

We regret the contradictory statements in the previous manuscript. Exact subject and SM exclusion criteria have been added in sections 2.3.1. *Subject filtering based on imaging data* and 2.3.2. *Subject measure filtering*. The text has been changed to reflect that participants are excluded based on SMs only if they have more than 50% of the final 74 SMs missing. This resulted in only two subjects being removed.

- 12. The authors note a train-test cross validation procedure will be used. Please state what proportion of participants will make up the training dataset and what proportion will make up the testing dataset?**

The text has been updated (2.4.2. *Canonical Correlation Analysis*) to reflect that the train-test cross validation will utilize an 80/20 split as in the original study.

- 13. I agree with the authors that using a developmental cohort is a strength as it allows assessment of whether Smith et al.'s original findings generalise to other groups. However, is it possible that because participants are likely developing at different rates, there is a heightened level of variation in the ABCD data and could this bias the results? Could this be examined in some way?**

We are pleased to hear the reviewer also sees the developmental cohort as a strength of our manuscript. We agree there is a possibility of greater variance in this dataset which should provide a strong test of the generalizability of the positive-negative axis finding. The degree of variance in the sample could potentially be systematically varied and tested for bias; however, we feel this is beyond the scope of the current manuscript. It should be noted that ABCD is a longitudinal sample with scans collected every 2 years for the next 10 years. It will be interesting to repeat the analysis on the same group of participants after they reached young adulthood. To facilitate this we made our data preparation/analysis pipeline as turn-key as possible so that researchers can re-run the pipeline on subsequent ABCD releases and investigate the CCA and positive-negative axis over time.

- 14. The final paragraph (page 7, line 7) can use some restructuring to make it clearer. First, break it into two paragraphs, one focusing on parcellations and the other based on how functional connectivity (FC) is calculated. Second, make it clear how the ICA in the ABCD data is being**

done (see comment 1). Finally, for the paragraph on FC calculation I was left a little confused as it made it sound like the main analyses (i.e. the computational and conceptual replication) are using different FC calculations, when they are in fact using the same one. Can this be rewritten so it is much clearer that this secondary analysis is being done using a simple Pearson correlation?

This has been clarified by restructuring the paper and removing the discussion of a computational replication (this discussion has been moved to a pre-print, citation below). The ABCD group-ICA is discussed in detail in the section 2.3.3. *Group-ICA of ABCD resting-state fMRI data* of the rewritten manuscript.

Goyal N, Moraczewski D, Thomas AG. 2020 Computationally replicating the Smith et al. (2015) positive-negative mode linking functional connectivity and subject measures. *bioRxiv*, 2020.04.23.058313. (<http://dx.doi.org/10.1101/2020.04.23.058313>)

15. On page 3 line 22 and page 4 line 15, it says participants aged between 9-10 are used but on page 5, line 21 it says participants are aged between 9-11. Could this discrepancy be fixed please.

We regret this error. All participants in ABCD Release 2.0.1 are between the ages of 9 and 10 years. This discrepancy has been fixed.

16. The abstract needs to be slightly reworked in my opinion, primarily to make it clear the focus is on networks as currently there is only one sentence which mentions networks. The term “functional brain activity” is used a lot, but to me this just sounds like a standard fMRI contrast. I would revise it to “functional brain connectivity”, “functional connectivity”, “functional brain network activity” or some variant thereof. In light of this I would change the order of the opening sentences of the abstract so it flows more logically. Something like “Understanding how brain functional activity is linked to human behaviour is a major goal of neuroscience. By modelling brain activity data as networks, known as functional connectivity, researchers can leverage the mathematical tools of graph theory to probe these relationships. Numerous studies have investigated the relationship between functional connectivity and.....”. Something like that.

We agree with the reviewer’s suggestion. The abstract has been entirely rewritten and this change incorporated.

17. The abstract should also mention the effect of parcellation is being examined as it is a big novelty of the study!

Although our first submission proposed exploring the effects of different parcellation on the CCA analysis, we have removed this aspect of our study. See our summary comments at the top of this document for details and rationale.

18. *Following from this, in the introduction of the paper itself I would include a brief discussion of how a network is constructed, much like the abstract does, to just help orient the reader a bit better (especially because in the introduction the idea of networks is already bought up).*

For brevity, we have removed this section of the paper and instead focus the introduction on the original study and its significance, and our replication and its potential implications should it be successful. We cite relevant literature for readers wanting to learn more about constructing brain networks.

19. *On page 3, line 55 the term “graph network” is used. These words are basically synonymous with each other, just use network.*

We thank the reviewer for this note. The term "graph network" does not appear in the rewritten manuscript.

20. *In the third paragraph in the section on network neuroscience where parcellations are discussed for the first time (page 3), this opportunity should be used to discuss (even if just one sentence) that different parcellations can get different results (see Zalesky et al., 2010; Sala-Llonch et al., 2019).*

We have removed the investigation of parcellation from this replication, but we have cited both of these papers and discussed the importance of parcellation in the introduction. We regret that time and resources do not allow the current set of authors to properly explore the effects of parcellation on this particular research question; however, our group intends to return to the topic in the future. Our shared code should also make it easier for other authors to explore this question with respect to the positive-negative axis finding and the ABCD dataset. See our summary comments at the top of this document for details and rationale.

21. *Relating to my previous comment on page 4 line 47, it is mentioned in parenthesis that the type of parcellation can affect the result. This comes across as a tangential point when it really deserves more attention to help properly establish a rationale for why a different parcellation is being considered. Also in this part there is a hyperlink to Sala-Llonch et als 2019 study. Please make sure to cite them properly, they did a lot of hard work and deserve a full citation!*

We regret this unconventional reference, we have now included a full citation to Sala-Llonch et al., 2019 in the introduction.

The following comments are more suggestions and I am happy for the authors to state that addressing these is beyond the scope of the paper, but I would suggest the authors give it some consideration as I think it might help the paper be more impactful:

22. *A novel aspect of this replication is the proposal to use an a priori parcellation (Gordon) over one derived from ICA. Given the effect of parcellation on network properties is well known, I would also suggest that the authors use this opportunity to examine the effects of other*

parcellations (for ease, this can just be restricted to the HCP data as I imagine there are subtleties in applying an adult parcellation of adolescent brains). For example the recently defined Schaefer parcellation (Schaefer et al., 2018) has received great interest as of late, and is also defined at a number of different nodal resolutions which would potentially allow for some extra exploration of how the number of nodes may affect the positive-negative axis (all the code/parcellations are freely available). However, at a minimum I strongly encourage the use of at least one other a priori parcellation as this would help make stronger statements about the effect of defining a parcellation a priori or a posteriori.

We share the reviewer's interest in the effects of parcellation, but we have chosen to remove this exploration from the current replication and hope to return to it in a subsequent paper. See our summary comments at the top of this document for details and rationale.

- 23. Additionally, to further drive home points about parcellation usage, including an anatomical parcellation (even if it is just the standard 34 cortical node one you get from Freesurfer) and seeing how that performs might be valuable. While this result will likely show weaker correlations than using a functional parcellation, many papers do analyse fMRI data using an anatomical parcellation so this might be a good opportunity to add more nails to that coffin.*

We agree with the reviewer on this point but have opted to set aside these nails for another paper. See our summary comments at the top of this document for details and rationale.

- 24. As a final general comment, the authors may wish to read Wang et al., 2018 (available as a preprint currently), who has written an entire paper on the use of CCA in neuroimaging/biomedicine. It may offer some useful commentary on interpreting the CCA or dealing with this kind of data.*

We thank the reviewer for this suggestion. The Wang 2018 paper was very helpful in interpretation of the original study's CCA results. We have cited it in the revised manuscript.

Reviewer 2

1. Stage 1 Primary Criterion #1: Have the authors provided a sufficiently clear and detailed description of the methods for another researcher to closely replicate the proposed experimental procedures and analysis pipeline, and to prevent undisclosed flexibility in the experimental procedures or analysis pipeline?

a. No. The Methods section is insufficient and does not allow for a full evaluation of the plans for this registered replication. The methods should include a clear description of all data and analyses to be performed; ascertaining many of the analytic details requires reading the original study, which shouldn't be necessary. Some of these details are found in other sections of the Introduction, which makes for a difficult reading experience.

We apologize for the missing details - the rewritten manuscript now addresses all of these aspects in detail in their own respective sections. Section 2.1. *Participants* discusses the ABCD sample demographics; 2.2. *Data Acquisition* provides scanning parameters, and discusses the types of subject measures and how they were collected; 2.3. *Data Preprocessing* covers in depth how the final subject sample and final subject measure set were selected; and section 2.4. *Data Analysis* reviews how the connectome and subject measure data were prepared for the CCA analysis, and what statistical analyses were performed in addition to the CCA.

b. In terms of the datasets, it's great that the authors have provided the specific NDA collection numbers, as this provides a link to the exact datasets analyzed. However, these cannot be accessed without an executed NDA data use agreement, so most readers won't be able to simply look up the details of these collections. The manuscript did not include any other description of the data, not even sample size. The original Smith et al. (2015) study included 461 participants from the HCP 500 subject release.

We regret these omissions. A table with details about the subjects (specific sample size, age, gender, race, sibship breakdowns) in ABCD collection 3165 has been added to section 2.1. *Participants*.

c. In the proposed replication, no description is provided as to what inclusion/exclusion criteria will be applied to the HCP 500, HCP 1200, and ABCD release 2.0 datasets, particularly those related to image quality metrics or data completeness.

All discussion of a computational replication with HCP 500/1200 data has been removed from this study to narrow its focus. Details of the computational replication have been moved to a preprint, cited below.

The inclusion/exclusion criteria applied to the ABCD subjects and subject measures are now outlined in sections 2.3.1. *Subject filtering based on imaging data* and 2.3.2. *Subject*

measure filtering. Exact numbers of subjects excluded by the various exclusion criteria are provided. Our ultimate reproducible workflow will provide a list of subjects included.

Goyal N, Moraczewski D, Thomas AG. 2020 *Computationally replicating the Smith et al. (2015) positive-negative mode linking functional connectivity and subject measures*. *bioRxiv*, 2020.04.23.058313. (<http://doi.org/10.1101/2020.04.23.058313>)

- d. *No description is provided of the imaging parameters, no description is provided of the behavioral measures. An Appendix A was referenced, but was not included in the manuscript submission. These details are critically important to evaluating the replication.***

We regret these missing details. The omission of Appendix A was due to confusion regarding what portion of the paper should be included for a results blind submission. The information about imaging parameters is now located in section 2.2.1. *Imaging data*, and behavioral measures in 2.2.2 *Subject measure data*. Appendix A is now included and has been updated to include the subject measures that are used in our study (either as confounds, or inputs to the analysis). Supplementary File 1 has been included in our submission which lists all 64,000 original SMs, and their reason for exclusion based on the quantitative exclusion criteria. Supplementary File 2 contains the one-to-one matching of the available ABCD SMs to HCP SMs used by Smith et al.

- e. *In terms of the analyses, the parcellations and CCA analyses are loosely described elsewhere, so getting a sense of what the authors will actually do requires pulling bits of text from different places in the Introduction. A clear description of the overall workflow, presented solely in the Methods and not the Introduction, would make the reader's experience much more enjoyable, and are needed for a complete evaluation of this work. Removing these analytic details from the Introduction will allow for a clearer presentation of the motivation and rationale (see below).***

We regret the poor structure of the initial submission - our rewritten manuscript now discusses methodology only in section 2. *Methods*. Motivation and rationale for the study are discussed in the introduction section 1.3. *Conceptual Replication*.

- 2. *Stage 1 Primary Criterion #2: Does the manuscript describe a sufficiently valid (i.e. close) and robust (e.g. statistically powerful) replication of the original study methods and rationale to provide an indication of replicability?***

- a. *No. The authors seek to replicate the results of Smith et al. (2015), both in the same dataset and an independent dataset, while testing for the effect of parcellation approach. While it's commendable that they're investigating the effects of parcellation and dataset on the analysis, the two datasets are so different from one another that generalization will be difficult to interpret.***

We thank the reviewer for this important point. We note that the investigation of parcellation has now been removed as discussed in our summary comments at the beginning of this document.

The two datasets are indeed very different and in the revised manuscript we have tried to highlight them as such. To address the need for a precise and interpretable replication, we have also posted a separate preprint with a more precise computational replication of the original study as well an expanded analysis with the 1200 subjects release of the HCP study.

To explore how generalizable these results are, unfortunately there are relatively few datasets available that have the requisite size and scope to successfully conduct a CCA analysis of this type. We chose the ABCD dataset because it had these requisites. In addition, because ABCD is longitudinal over 10 years, it also has the exciting potential for future studies to explore changes in the positive-negative axis as the ABCD subjects become closer in age to the HCP subjects.

We agree that if the Smith et al. results fail to replicate in the ABCD dataset, there will be many potential reasons including differences in the subject population, scanner and/or site differences, and scan length among many others. Exploring the necessary and sufficient conditions for observing the results reported in Smith et al. would then be the subject of future work from our group and others.

- b. Three major factors that are not investigated in this design are participant age (see below), study site (esp. scanner), and behavioral metrics (insufficient details were provided on these). (As an aside, WUSTL is the HCP site, but is also an ABCD site, so there is a missed opportunity to analyze both HCP and ABCD data acquired at that site.) Other factors worth looking into include preprocessing pipeline, acquisition parameters, and sample demographics (beyond age), none of which were mentioned.*

We agree that it would be very interesting to explore how these factors impact the CCA results as compared to the original study.

Age is included as a confound regressor, but obviously there is a large and unavoidable age difference between the two datasets. Please see our response to Reviewer 1, comment 13 and our responses to comment 3 below for further discussion of this topic. We have added more discussion of study site to the manuscript -- it will be treated as Smith et al. treated processing pipeline. We have also included more detail on our efforts to match the behavioral metrics across the two studies in sections 2.3.

We were not able to determine to what extent the scanner and facility at WUSTL had changed in the ~6 year period between the HCP and ABCD data collections thus we did not include this analysis in our study. However, we will design our data

preparation/analysis pipeline to be turn-key for researchers to explore all of these factors longitudinally, as subsequent ABCD datasets are released. To facilitate this all of our tools and data appropriate for sharing will be made available on the project's Open Science Foundation page (<https://osf.io/qnp62/>).

- c. *Finally, it's unclear what would constitute a successful/failed replication in any of the analysis, which is a critical aspect of the plan that was not addressed.*

We apologize for this omission. The introduction has been revised and now has an explicit definition for a successful replication.

- d. *Sample size and power were not addressed.*

We have updated the manuscript to include the final sample size (7,810 subjects, and 74 subject measures). Although we have not conducted a formal power analysis, this is an order of magnitude more subjects than the original study, thus we feel we have adequate power to detect the results reported in Smith et al. There is a possibility that the shorter scan length in ABCD will reduce our ability to estimate connectivity networks but this is not obviously addressable with a power analysis.

3. *Stage 1 Secondary Criterion #1: Are the proposed hypotheses logical and supported by a clear and persuasive scientific rationale?*

- a. *No. I'll start with my concerns regarding the organization of the manuscript, as these are coupled with my concerns regarding presentation of the scientific rationale for the proposed replication. Overall, although it was brief, I found the manuscript fairly challenging to read, and revising the organization and presentation of various pieces are likely to significantly improve the manuscript.*

We apologize for the poor organization of the original submission. The entire manuscript has been rewritten and restructured for proper organization.

- b. *Much of the Introduction is a broad and diffuse summary of connectomics, which could be removed in favor of a well-organized description of the rationale and motivation for the proposed replication.*

Thank you for this suggestion. We have removed the discussion of connectomics and instead focused the introduction on the findings of Smith et al., the importance of their findings, and the rationale/motivation for our replication.

- c. *The initial citation of 7 studies relating brain and behavior seems to be hand-picked and does not have a clear purpose. The next paragraph is a summary of the study to be replicated (Smith et al., 2015), but the motivation for the replication presented here is simply to assess the robustness of the original finding. This is not a compelling*

argument, particularly in light of the age differences between the HCP and ABCD samples.

The introduction has been completely rewritten, and the particular sentence in question has been removed. We have also clarified our motivation for the replication to reflect that our goal is not only to assess the presence of this landmark finding in an independent, pre-adolescent dataset. The manuscript reflects that our motivation is to replicate the positive-negative axis in a heterogeneous dataset to demonstrate that this phenomenon is independent of such differences (see section 1.3.1. *Replication overview* for full details); we believe that this will enable researchers to establish a signature neurotypical relationship between connectivity and phenotype, which could then be applied to a variety of clinical populations. Most importantly, finding the positive-negative axis in the ABCD dataset would enable researchers to study it longitudinally over subsequent ABCD releases for the next 10 years. In doing so, researchers could explore how the positive-negative axis changes with regard to age, behavioral characteristics, mental and physical health outcomes, education, demographics, and environment, possibly even determine which factors form the basis for the positive-negative axis.

- d. From here, the section on “Network neuroscience” backtracks to a broad review of fMRI, with an abrupt transition to parcellations and graph theory. This text isn’t needed, and it detracts from what should be the purpose of the introduction: to set up the purpose of the replication.***

We agree and apologize for the unnecessary text. It has been removed and the introduction is now focused on the purpose of the replication.

- e. The final paragraph of section 2 is briefly describes the HCP and ABCD projects, but this seems out of place and insufficient. HCP inclusion of “sibships” is mentioned (without a definition), without similarly noting the inclusion of twin data in the ABCD sample.***

We regret this omission. The original paragraph has been removed. The discussion of the HCP project has been kept separate from the ABCD dataset, and is discussed in the introduction. We have updated the text and included sibling data in section 2.1. *Participants* with a full breakdown of MZ and DZ twins, regular siblings, and single children. The term sibships is defined in that section as well.

- f. From there, the introduction swings back to the original Smith et al. (2015) study and loosely describes the original methodology. It would be helpful if this section also included a summary of the significance of this finding. When the original Smith et al. (2015) study is introduced, identifying the significance of this finding will help in articulating the significance of the replication. I’m not someone who thinks that manuscripts should be rejected for “not being significant enough”, but I do think that authors should strive to translate data-driven findings into a statement that summarizes***

their significance or implications. This doesn't appear to be addressed, but to be fair, I'm not certain that it was addressed in the original publication (likely a function of the Brief Communication format). It's remarkable that "a single highly significant CCA mode that relates functional connectomes to subject measures" was observed, and that "nearly all the positively correlated SMs are commonly considered as positive personal qualities or indicators... and all negatively correlated SMs relate to negative traits". But beyond this finding itself, it's not clear how the result informs our understanding of human functional brain organization. The manuscript then states that the motivation of the replication is to address the generalizability of the results with respect to the brain parcellation and the study sample (see above for comments on the challenges of addressing generalizability with this design).

We agree with the reviewer, we have expanded on the importance of the original author's findings and the potential implication of our replication if successful. We have also added a citation to the accompanying paper (Holmes & Yeo, 2015, <https://doi.org/10.1038/nn.4145>) published in the same issue of Nature Neuroscience which provides more details on the significance of the original finding.

- g. It is stated that the HCP sample is representative of the US population, but this term should be avoided, both with respect to HCP and ABCD samples (please see Compton, Dowling, and Garavan, JAMA Pediatr, DOI: 10.1001/jamapediatrics.2019.2081).*

Thank you for these sources. We have removed the claim that these samples are truly representative. We will also include this as a limitation in the discussion section of the paper, citing the above reference.

- h. Parcellation choice is an important aspect for this replication of Smith et al. (2015), and I agree with the authors that multiple parcellation approaches should be tested to address generalizability of the findings. However, a replication and extension that leverages both HCP and ABCD samples should include clear age-related questions, as this is a critical difference between these two study samples.*

We agree that age-related changes in brain-behavior relationships is the critical question and that the ABCD dataset will be invaluable to these investigations. However, since we are only examining the first time point of this dataset, we have minimal variance in participant age. Given inter-individual variability and the minimal slice of developmental trajectory provided in this dataset release, we chose to not examine the effect of age in this analysis; however, our tools will enable us and other researchers to examine this effect as new timepoints are released. In addition, please see comment 2j regarding the age-related questions.

- i. Importantly, the ABCD release 2.0 baseline data includes youth aged 9-10 years old (the manuscript should be revised throughout to reflect that these are pre-adolescent, not adolescent, participants).*

We regret this error. The manuscript has been corrected to reflect this.

- j. Extending this replication of Smith et al. (2015) to ABCD data introduces a lifespan question that asks whether the observed feature of brain organization is present prior to the neurodevelopmental changes that occur during adolescence and early adulthood. In this respect, the manuscript does not at all address the age-related relevance of extending the analysis from HCP to ABCD samples. The authors “hypothesized that the single CCA mode and positive-negative axis found in the original study would successfully replicate in the ABCD dataset, regardless of the parcellation and subject measures used”, but what is the justification for this hypothesis? What evidence suggests that this single CCA mode and positive-negative axis is present prior to adolescence?***

We apologize for the lack of justification in the initial submission - since the Smith et al. study has not been replicated in an independent dataset to date, we cannot definitively comment on whether or not the positive-negative axis will appear. However, a publication by Xia et al. 2018 which used neuroimaging data of youth (8-22 years old), found positively correlated patterns of functional connectivity and psychiatric symptoms across 4 dimensions of psychopathology, with each dimension being associated with a distinct pattern of abnormal connectivity. To account for potential confounds, the authors controlled for age, sex, race, and motion, and they also found that these findings replicated in an independent dataset. Their study suggests that it is possible to conduct a CCA-type analysis on highly heterogeneous youth data and obtain interpretable results, and that there appears to be a common neurobiological mechanism underlying vulnerability to a wide range of psychiatric symptoms echoing the positive-negative axis finding. Given their findings, we feel that the ABCD dataset is appropriate for the current study despite the age difference and heterogeneity compared to HCP. This information is now included in the manuscript in section 1.3.1. *Replication overview*.

Should the positive-negative axis fail to replicate in the ABCD sample, we still see this dataset as the best choice for replication since researchers can conduct the CCA analysis longitudinally as subsequent ABCD datasets are released; researchers could then investigate when the positive-negative axis appears and study the effects of age on the phenomena.

Xia CH, Ma Z, Ciric R, et al. Linked dimensions of psychopathology and connectivity in functional brain networks. Nat Commun. 2018;9(1):3003. Published 2018 Aug 1. <http://doi.org/10.1038/s41467-018-05317-y>

- k. Related to motivation and rationale, I found the presentation of the analytic strategy (page 5) to be less than compelling. It comes across as if the original choice of parcellation is based on convenience: the authors will employ the same parcellation as Smith et al. (2015) for the HCP data, and then use the Gordon parcellation for the***

ABCD data. As a reader, convenience is never a sufficient justification, so this raises concerns. But then the manuscript points to a figure (although not spelled out in the text) that indicates that multiple analyses will be performed such that both parcellations will be employed on both samples. This is an awkward presentation of a critical aspect of the replication. Moreover, the figure includes the HCP 500 and ABCD 2.0 analyses, but not the HCP 1200 analysis.

We apologize for the poor presentation of the analytic strategy - we have rewritten the manuscript so that the strategy is clearly defined. In addition, since analyzing the effect of parcellation is no longer an aspect of this study, this figure and associated text have been removed. The only parcellation under consideration in the revised study is the 200-dimension group-ICA of the ABCD data to be consistent with the original study.

1. *I am highly enthusiastic about registered replications, but I found this Stage 1 manuscript to be quite difficult to read and review. Replication of the Smith et al. (2015) finding is sound idea, and I applaud the first author for taking on such an ambitious project and making use of both HCP and ABCD datasets. However, the execution of this idea requires significant additional justification and will require additional consideration of age-related differences that are likely to impact generalizability. Overall, there are a number of details that need to be clarified and the manuscript requires extensive rewriting and additional oversight.*

Thank you for your feedback, it was extremely helpful when revising our manuscript.

4. *Stage 1 Secondary Criterion #2: Are the proposed procedures and analysis pipeline methodologically sound?*

- a. *Insufficient details provided to fully evaluate.*

We apologize for the lack of detail - discussion of the procedures/pipeline have been rewritten to be explicit and concise. Please see section 2.3. *Methods*.

5. *Stage 1 Secondary Criterion #3: Have the authors considered sufficient outcome-neutral conditions (e.g. absence of floor or ceiling effects; positive controls; other quality checks) for ensuring that the results obtained are able to test the stated hypotheses?*

- a. *No. It's unclear what would constitute a successful/failed replication in any of the analysis. Moreover, while parcellation and dataset will be tested, other important factors will not (e.g., age, study site, behavioral metrics, imaging parameters, preprocessing, etc.), which will likely contribute to ambiguity in results.*

We apologize for the ambiguity. We have now explicitly defined what constitutes a successful replication. In addition we discuss post-hoc analyses which will be conducted. As discussed in responses 2a & 2b above, we recognize the many differences between

these two datasets, but do not feel that we are able to comprehensively explore all of the potential source of variance. We leave this as an avenue for future work should the results fail to replicate. We have, however, outlined how we will attempt to control for these variables as much as possible. As subsequent ABCD datasets are released, researchers can utilize our pipeline to explore questions about how age, study sites, behavioral metrics, etc. impact the results.

6. Do you have any concerns about the proposed statistical analyses in this study? If so, please specify them explicitly in your report.

None.

7. It is a condition of publication that authors make their supporting data, code and materials available. Is it clear from reading the paper how the authors will make these available?

a. The link to the GitHub repository should be included.

A statement with a link to the project's *Open Science Foundation* page (<https://osf.io/qnp62/>) has been added to section 2.5. *Code availability*. All code and relevant data that we are allowed to release will be made available in this repository.

8. Comment on the Title

a. Please remove the second comma from the title (and add an Oxford comma after "demographics"). Also, the citation of the original study should not be possessive (i.e., it should be *Smith et al. (2015)* instead of *Smith et al.'s*).

The title has been corrected.

Reviewer 3

From section 4.3: I think it would be good to be slightly more precise about how the CCA output(s) will be compared against the original results - I would say it is not just important to show whether one or more CCA components are found, but how similar they are in terms of the functional networks and non-imaging-variables identified.

We agree with the reviewer that a direct comparison between the original HCP and ABCD results could provide a clearer interpretation of our results. Our goal in the current study is a conceptual rather than a quantitative replication. One important point is that the HCP and ABCD datasets each have their own unique group-derived 200-dimension ICA parcellations. Thus, a correlation between functional network-projected CCA modes from HCP and ABCD, respectively, would not be interpretable since the parcellations for each dataset are different. Further, while there is some overlap in SMs, the HCP and ABCD CCAs have differing SMs.

In order to facilitate our goal of conceptual replication, we have incorporated post-hoc analyses for both the SM and connectivity data. For the SM results, mapping the correlation between the CCA and each SM seeks to conceptually replicate the relationship between the SMs and the mode. For the connectivity data, we submit our group-averaged parcellation to a hierarchical clustering algorithm to examine how our parcellation maps onto known functional networks and those found in Smith et al. 2015. We also compare the 30 connections that are most strongly associated with the CCA mode (as in Figure 2A in Smith et al.) across the two datasets. We have added text to section 2.4.3 Post hoc analyses to reflect these analyses.

The paper is too vague about what data was used from ABCD and what preprocessing was applied.

We apologize for the lack of details in the original submission. The revised manuscript now discusses the how we quantitatively/qualitatively selected which ABCD data to use (section 2.3.) and the preprocessing applied to the scanning data (see beginning of section 2.3. and section 2.3.3.)

1. ***Were structured artefacts removed?***

In accordance with the original study, we remove structural artifacts using ICA+FIX. This detail has been added to section 2.3.3.

2. ***Ideally to best match HCP preprocessing, ICA cleanup should be used, or an alternative that has been validated to be similarly effective.***

We have updated the manuscript to reflect that we apply ICA+FIX, to be consistent with the original study.

3. ***Was global signal regressed out? Hopefully not.***

4. **Global signal was not regressed out, this is now made explicit in the text. A full explanation of the pre-processing is now included at the beginning of section 2.3. Data Preprocessing.**

5. *The implication is that this was all surface/CIFTI-based analysis but this needs to be described more explicitly.*

The reviewer is correct that all analysis, after surface projection during preprocessing, was done in CIFTI space. This has been clarified in the text under section 2.3 - Data Preprocessing.

Appendix C

Responses to Reviewers (Minor Revisions)

Reviewer: 1

1. *Super minor thing: On page 4 of the manuscript, lines 43-49, these two sentences start the exact same way ("Prior to filtering"). In the spirit of good English please avoid this repetition.*

This has been corrected, thank you for bringing it to our attention.

Reviewer: 2

1. *Manuscript page 3, line 17, I suggest specifying that "finding the positive-negative axis in the baseline ABCD dataset would enable..." (since you are pointing to follow-up work with other time points)*
2. *Table 1, replace headers with "Sex" instead of "Gender" and "Race/Ethnicity" instead of "Race"*

Both changes have been made, thank you for bringing them to our attention.

Reviewer: 3

- 1. One concern is whether ABCD is close enough (particularly wrt age) to HCP for this work to be considered a *replication* study. I'm on the fence on that. I think it just depends on your definition of "replication". Maybe this should be called a "generalisation" study?***

We agree with the reviewer that the definition of “replication” required additional clarification. We have added the following clarifying text to the beginning of section **1.3. Conceptual replication in an independent dataset** (the section has also been renamed from *Replication using an independent dataset*):

“This is not a replication in the sense that we seek to find results identical to those of the original authors (an exact computational replication of the original study is documented in our preprint Goyal et al., 2020). Rather, we apply the methodology of Smith et al. 2015 to the ABCD dataset to determine if there exists a strong correlation between connectomes and SMs and dominant CCA mode that explains a significant amount of covariance within a dataset with very different subject and scanner characteristics.”

Goyal N, Moraczewski D, Thomas AG. 2020 Computationally replicating the Smith et al. (2015) positive-negative mode linking functional connectivity and subject measures. bioRxiv , 2020.04.23.058313. (doi:10.1101/2020.04.23.058313)

- 2. My other concern is: if the idea is to show that one and only one significant CCA mode is found, there is the problem of increased statistical sensitivity by using much higher number of subjects in this study. Obviously finding more than one significant mode could be just a function of increased N, and so that would not be "failure". A more subtle issue is that in PCA and CCA there can be arbitrary rotation amongst components, so that if more than one mode were found, the original result might validly be mixed up across them***

Thank you for bringing this to our attention - we agree that it is possible for multiple significant modes to occur given the size of the dataset, and have added the following text and citation to Wang et al. 2020 to section **1.3.1. Replication Overview**:

“Although we are particularly interested in finding a single dominant CCA mode, given the large sample size of the ABCD dataset it is possible that multiple significant modes exist. The appearance of these modes would not necessarily indicate a failure to replicate, but would warrant a careful analysis of the results since in PCA/CCA there can be an arbitrary rotation amongst components that can spread the original result validly across the multiple significant modes (Wang et al., 2020).”

Wang H-T, Smallwood J, Mourao-Miranda J, Xia CH, Satterthwaite TD, Bassett DS, Bzdok D. 2020 Finding the needle in a high-dimensional haystack: Canonical correlation analysis for neuroscientists. Neuroimage , 116745. (doi:10.1016/j.neuroimage.2020.116745)

Appendix D

Dear Professor Chambers,

We are pleased and grateful that you and the reviewers felt that our Stage 1 replication was suitable for an in-principle acceptance. We have implemented the accepted protocol and enclosed is our Stage 2 submission. Please see our cover letter for further details.

Best Regards,

Dustin Moraczewski

Appendix E

Dear Professor Chambers,

We pleased and grateful that you and the reviewers felt that our Stage 2 replication was suitable for a conditional acceptance. We have revised our manuscript according to the reviewer's suggestions. Please see our cover letter for further details.

Best Regards,

Dustin Moraczewski